# Activated NK cells cause placental dysfunction and miscarriages in fetal alloimmune thrombocytopenia

Issaka Yougbaré[1,2,3], Wei-She Tai[1,2], Darko Zdravic[1,2,3,4], Brigitta Elaine Oswald[1,2,3,4], Sean Lang[1,2,3,4], Guangheng Zhu[1,2], Howard Leong-Poi[5], Dawei Qu[6], Lisa Yu[6], Caroline Dunk[6], Jianhong Zhang[6], John G. Sled[6,7,8], Stephen J. Lye[6,9], Jelena Brkić[10], Chun Peng[10], Petter Höglund[11], B. Anne Croy [12], S. Lee Adamson[6,9,13], Xiao-Yan Wen[2,13], Duncan J. Stewart[14], John Freedman[1,2,4,15] & Heyu Ni[1,2,3,4,13,15]

Miscarriage and intrauterine growth restriction (IUGR) are devastating complications in fetal/neonatal alloimmune thrombocytopenia (FNAIT). We previously reported the mechanisms for bleeding diatheses, but it is unknown whether placental, decidual immune cells or other abnormalities at the maternal–fetal interface contribute to FNAIT. Here we show that maternal immune responses to fetal platelet antigens cause miscarriage and IUGR that are associated with vascular and immune pathologies in murine FNAIT models. Uterine natural killer (uNK) cell recruitment and survival beyond mid-gestation lead to elevated NKp46 and CD107 expression, perforin release and trophoblast apoptosis. Depletion of NK cells restores normal spiral artery remodeling and placental function, prevents miscarriage, and rescues hemorrhage in neonates. Blockade of NK activation receptors (NKp46, FcγRIIIa) also rescues pregnancy loss. These findings shed light on uNK antibody-dependent cell-mediated cytotoxicity of invasive trophoblasts as a pathological mechanism in FNAIT, and suggest that anti-NK cell therapies may prevent immune-mediated pregnancy loss and ameliorate FNAIT.

[1] Toronto Platelet Immunobiology Group, Keenan Research Centre for Biomedical Science, St. Michael's Hospital, Toronto, ON, Canada M5B 1W8. [2] Department of Laboratory Medicine, Keenan Research Centre for Biomedical Science, St. Michael's Hospital, Toronto, ON, Canada M5B 1W8. [3] Canadian Blood Services, Toronto, ON, Canada K1G 4J5. [4] Department of Laboratory Medicine and Pathobiology, University of Toronto, Toronto, ON, Canada M5S 1A8. [5] Division of Cardiology, St. Michael's Hospital, University of Toronto, Toronto, ON, Canada M5B 1W8. [6] Lunenfeld-Tanenbaum Research Institute, Mount Sinai Hospital, Toronto, ON, Canada M5G 1X5. [7] Research Institute, Hospital for Sick Children, Toronto, ON, Canada M5G 0A4. [8] Department of Medical Biophysics, University of Toronto, Toronto, ON, Canada M5G 1E2. [9] Department of Obstetrics and Gynecology, University of Toronto, Toronto, ON, Canada M5G 1E2. [10] Department of Biology, York University, Toronto, ON, Canada M3J 1P3. [11] Department of Medicine Huddinge, Center for Hematology and Regenerative Medicine (HERM), Karolinska Institutet, Stockholm 141 86, Sweden. [12] Department of Biomedical and Molecular Sciences, Queen's University, Kingston, ON, Canada K7L 3N6. [13] Department of Physiology, University of Toronto, Toronto, ON, Canada M5S 1A8. [14] Ottawa Hospital Research Institute, Ottawa, ON, Canada K1H 8L6. [15] Department of Medicine, University of Toronto, Toronto, ON, Canada M5S 1A8. Correspondence and requests for materials should be addressed to H.N. (email: nih@smh.ca)

Fetal/neonatal alloimmune thrombocytopenia (FNAIT) is a life-threatening gestational disease characterized by maternal immune responses against fetal platelet antigens. FNAIT leads to fetal/neonatal platelet destruction, bleeding disorders ranging from mild cutaneous petechial to severe intracranial hemorrhages (ICH), and fetal or neonatal death[1–4]. Incompatibilities in gene polymorphisms between the mother and fetus initiate the immune response[3, 5]. A total of 36 alloantigens have been reported and approximately half are located on the extracellular domains of integrin β3 subunit[3, 4]. In Caucasians, 70–90% of reported cases are caused by human platelet antigen-1a, which is due to a gene polymorphism in residue 33 (L33P) in β3 subunit[3, 5]. Maternal antibodies generated during pregnancy cross the placenta and target paternally inherited antigens on platelets and other cell types, causing FNAIT[6–8]. We previously demonstrated that transplacental passage of maternal anti-β3 integrin antibodies impairs mouse fetal blood vessel development and causes bleeding particularly in fetal and neonatal brains[7, 9]. Prevalence of FNAIT is estimated at 0.5–1.5/1,000 liveborn neonates, but this number is inaccurate because it does not include miscarried fetuses that are inadequately documented[10, 11]. Some reports estimate that up to 30% of affected fetuses miscarry[12]. Mechanisms for in utero fetal death and for reported intrauterine growth restriction (IUGR) in FNAIT, however, are largely unknown[3, 13–15].

The most targeted antigen in FNAIT, β3 integrin, is not only expressed on platelets and endothelial cells, but also expressed on conceptus-derived trophoblast (placental) cells. Trophoblast αIIbβ3 and αVβ3 integrins are early contributors to blastocyst implantation and subsequent placental development including spiral artery (SA) remodeling[16–19]. Deficient SA remodeling is associated with pregnancy complications that include preeclampsia (a hypertensive syndrome of mid-late pregnancy), IUGR, and miscarriage[20–22]. β3 integrin-positive invasive trophoblast cells expressing paternally inherited alloantigens are reported to initiate immune responses through interactions with maternal decidual immune cells[23]. Whether paternal β3 integrin-positive trophoblast cells are recognized by the maternal immune system and whether their migration and functions in SA remodeling are impaired in FNAIT have not been explored[24, 25].

At early human and other mammalian implantation sites, natural killer (NK) cells are highly enriched, transient lymphocytes that promote decidualization, including immune tolerance and vascular development[26–29]. Unlike human peripheral NK (CD56[dim]), decidual NK (dNK) cells (CD56[bright]) are non-cytotoxic cells with angiogenic potential that appear to be essential for normal early decidual angiogenesis[30–32]. The importance of NK cells in successful pregnancy has been defined by studying pregnant mice devoid of NK cells, and by demonstrating angiocrine properties of uterine NK (uNK) cells from normal mice[33]. Mouse uNK cells are recruited in large numbers to the mesometrial decidua between days 6–11 of pregnancy[34, 35]. By mid-gestation (day 12), most mouse uNK cells have become senescent and cell numbers have declined[36]. Notably, switches in phenotypes and functions of d/uNK cells have been reported during both human and mouse gestation[37–39]; for example, in human pregnancy complications, different activating receptors (NKp30, NKp46, and Fc gamma receptor FcγRIIIa) and granule content (perforin and granzyme) are upregulated[40, 41].

Human and mouse d/uNK cells tightly control extravillous trophoblast (EVT/invasive) migration, making d/uNK and trophoblast cells partners during pregnancy[32, 42]. Human trophoblasts uniquely do not express human leukocyte antigen (HLA)-A or HLA-B but EVTs express HLA-C, E, and G, molecular ligands for NK cell allorecognition receptors[42, 43]. Perforin, released upon dNK activation, is a main mediator of cytotoxicity. During pregnancy, inflammation (e.g., induced by IL17-producing helper T cells (Th17), type 1 helper T cells (Th1), or lipopolysaccharide) may alter NK cell quiescence and lead to abnormal activation[44, 45]. Mounting evidence has linked NK cells to human and mouse reproductive failure particularly when the NK cells become adversely activated and mediate fetal demise by releasing perforin[46].

Placental pathologies are not well addressed in FNAIT, although a clinical study reported that IUGR and fetal demise occur as frequently as ICH[11]. Fetal loss may result from placental pathologies antecedent to ICH. Given that β3 integrin is expressed by trophoblasts, we hypothesized that in FNAIT maternal immune responses to fetal antigens may trigger IUGR and pregnancy loss, and that maternal anti-β3 integrin IgG may form immune complexes on trophoblast cells to create targeted binding sites for NK cell Fcγ receptors[41, 47]. These immune complexes would then trigger NK cell-mediated antibody-dependent cell-mediated cytotoxicity (ADCC), trophoblast cell death, and subsequent pregnancy failure. To our knowledge, this possibility has not been previously explored.

Here, using our murine FNAIT model[7, 48] and human trophoblast cell lines, we demonstrate that placental abnormalities lead to IUGR and miscarriage. This is likely due to uNK cell activation through NKp46 and perforin release causing trophoblast apoptosis. These uNK cell-mediated placental pathologies are prevented by anti-NK antibody treatments. Furthermore, these treatments reduce severity of bleeding in fetuses and neonates, demonstrating their potential for therapeutic applications.

## Results

**Maternal responses to fetal β3 integrin cause miscarriage and IUGR.** To mimic human FNAIT and exposure to β3 integrin antigens during gestation, $Itgb3^{-/-}$ females (referred to hereafter as $β3^{-/-}$) were transfused twice with wild-type (WT) platelets and then bred with WT males as previously reported[6, 7]. These $Itgb3^{-/-}$ females are referred to as "immune". Naive $β3^{-/-}$ bred with WT mice were used as the normal pregnancy control group since they do not develop detectable levels of anti-β3 integrin antibody in first pregnancy (studied here) or second pregnancy[48]. These females are designated "non-immune". Abnormal fetal development and death were frequent in immune females at E14.5 ± 1.5. Immune group fetuses had significantly lower body weights than gestational-day-matched control fetuses (Fig. 1a). This was accompanied by delayed organogenesis compared to fetuses from non-immune pregnancies. IUGR preceded fetal demise at E14.5 ± 1.5 in immune pregnant females. No deaths were observed among normally growing fetuses in non-immune pregnant $β3^{-/-}$ females (Table 1).

To address maternal immunity induced by platelet administration, splenocyte suspensions were immunophenotyped by flow cytometry. Splenomegaly was present in pregnant immune mice with increased $CD3^+CD4^+IL17^+$ cells (i.e., a Th17-polarized immune response) compared to non-immune $β3^{-/-}$ pregnant mice (Fig. 1b). As expected, immune $β3^{-/-}$ females bred with $β3^{-/-}$ males did not develop splenomegaly or miscarriage. To further demonstrate that maternal immunity specifically targets only fetuses carrying paternally derived antigen, immune $β3^{-/-}$ females were bred with $β3^{-/+}$ males. These females lost their heterozygote fetuses ($β3^{-/+}$; Supplementary Fig. 1) around E14.5. We further analyzed maternal plasma cytokines. IL12/23, IL17, and MCP-1 were detected in all pregnancies, and were significantly elevated in immune compared to non-immune pregnant females (Fig. 1c). In addition to Th17-polarized cytokines, angiokines were lower in both plasma and placental homogenates from immune mice. The plasma of immune pregnant mice had less placental growth factor (PGF), more

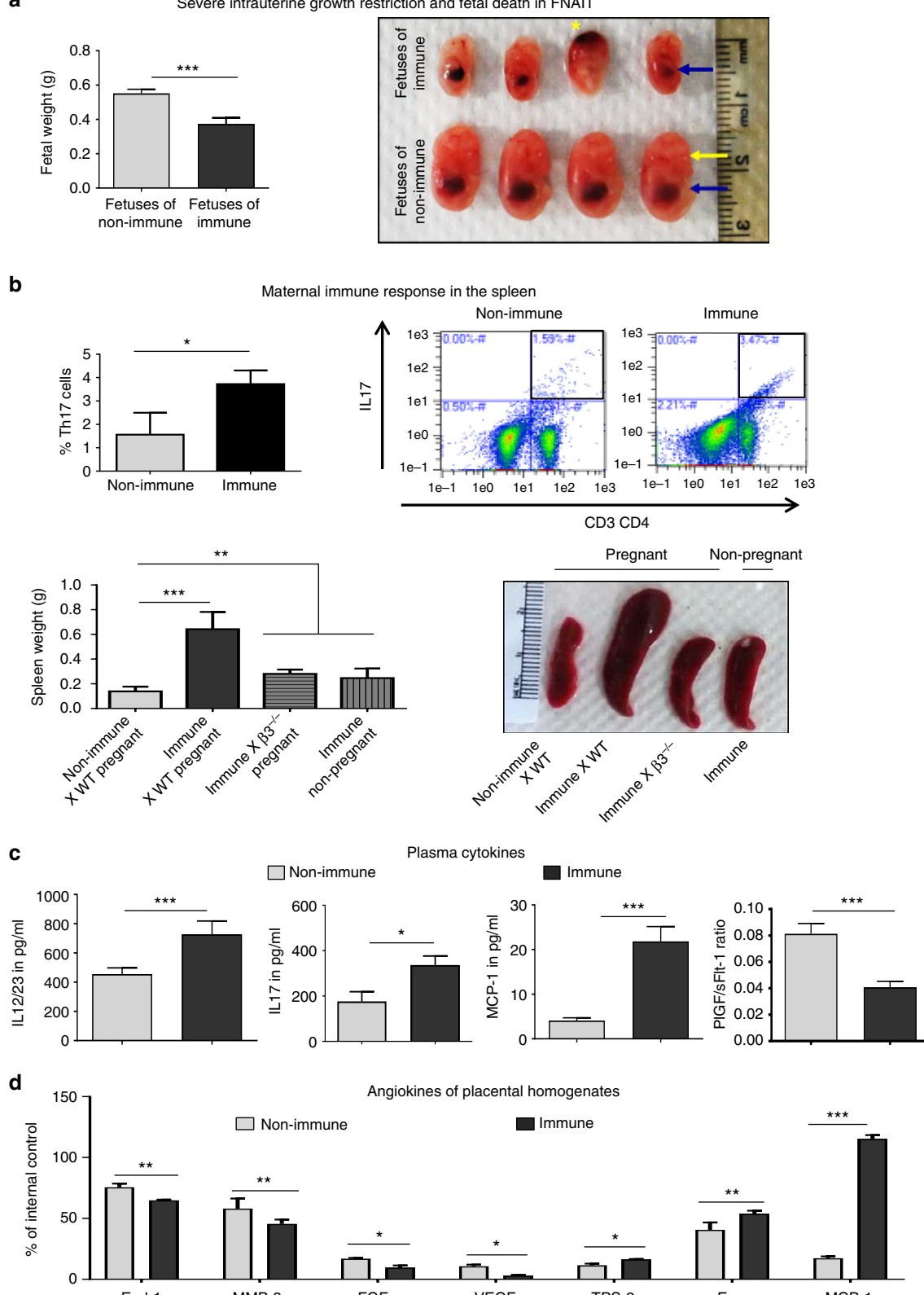

**Fig. 1** Maternal immune responses to fetal platelet β3 integrin cause IUGR and fetal death in FNAIT at E14.5. **a** Fetuses from immune mice had significantly impaired growth as revealed in the *top panel* by body weight and gross examination (liver = *blue arrow*, eye = *yellow arrow*, ICH = *yellow star*). **b** Immune mice developed a Th17 pro-inflammatory response (*top panels*) and splenomegaly (*bottom panels*) during pregnancy compared to non-immune pregnant mice. **c** Circulating Th17-polarized cytokines were also elevated in immune pregnant mice. Both plasma (**c**) and placental (**d**) angiogenic cytokines were significantly decreased in immune mice compared to non-immune pregnant mice. Data were collected from more than 30 pregnancies per group. *End-1* endothelin-1, *Eng* endoglin, *FGFa* fibroblast growth factor acidic, *MCP-1* monocyte chemoattractant protein-1, *MMP-3* matrix metalloproteinase, *TSP-2* thrombospondin-2, *VEFG* vascular endothelial growth factor, Unpaired Student's *t*-test. Mean ± SEM. *$p < 0.05$, **$p < 0.01$ and ***$p < 0.001$

| Table 1 Fetal survival decreases in FNAIT at mid-gestion | | | | |
| --- | --- | --- | --- | --- |
| | Fetal survival (%) and 95% confidence interval | | | |
| Conditions | E12.5 | E13.5 | E14.5 | E15.5 |
| Non-immune | 35/35 | 38/38 | 31/31 | 34/34 |
| | **(100%)** | **(100%)** | **(100%)** | **(100%)** |
| | 0 | 0 | 0 | 0 |
| Immune | 33/33 | 32/35 | 18/34 | 5/37 |
| | **(100%)** | **(92%)** | **(53%)** | **(14%)** |
| | 0 | 9 | 17 | 11 |

The majority of fetuses were found dead around E14.5 ± 1.5. Bold values represent survival percentage

soluble fms-like tyrosine kinase-1 (sFlt-1; Supplementary Fig. 2), and a significantly lower PGF/sFlt-1 ratio than plasma from non-immune pregnant mice (Fig. 1c). Since a lower than normal PGF/sFlt-1 ratio or supranormal soluble endoglin-1 levels contribute to the pathogenesis of preeclampsia in women[49] and of impaired placental vascularization in mice[50], additional placental angiogenic signaling molecules were investigated (Fig. 1d). In comparison to plasma from non-immune pregnancies, immune pregnant mice had significantly lower levels of pro-angiogenic molecules including endothelin-1, matrix metalloproteinase-3, fibroblast growth factor acidic, and vascular endothelial growth factor. In contrast, the potent anti-angiogenic factors endoglin and thrombospondin-2 and the pro-inflammatory cytokine monocyte chemoattractant protein-1 were increased.

**Impaired maternofetal placental exchange functions in FNAIT.** To assess fetal viability in vivo and placental hemodynamics, ultrasound was performed on anesthetized pregnant mice[7]. Living fetuses of immune mice displayed bradycardia compared to fetuses of non-immune mice (Fig. 2a). Chorionic plate capillaries in placentas of the latter conceptuses had large areas of Doppler velocity; these areas were smaller in placentas from immune pregnancies (Fig. 2a). To assess placental malperfusion, gross placental anatomy of viable fetuses from immune and non-immune mothers was compared ex vivo. Placentas of control fetuses appeared dark red, whereas placentas of fetuses from immune mothers were paler, suggesting ischemia (Fig. 2a). Slower Doppler velocity in the placentas from immune mothers suggests impaired placental vascularization or blood hyperviscosity. To discriminate between these, microcomputerized tomography (micro-CT) was conducted to visualize the arterial vasculature. Placentas from non-immune mothers showed normal arterial branching from the umbilical artery, and the injected yellow contrast agent showed typical distal fetal capillaries in the labyrinth (Fig. 2b). This was confirmed by stereomicroscopic pictographs of labyrinthine capillaries, as well as by quantification of immunofluorescent staining of paraffin-embedded placental sections stained by endothelial cell-reactive isolectin IB4 (Fig. 2b). In placentas from immune pregnancies, fetal capillary development was significantly reduced in the labyrinth (Fig. 2b), suggesting that deficient capillary development impairs placental exchange.

In vivo biotin transfusion studies were conducted to assess placental exchange function. Immediately after fetal echography on E14.5, biotin was injected intravenously into the pregnant dams. To test whether maternal endogenous biotin can cross the placenta and accumulate in fetal tissues, immunofluorescent staining was conducted[51]. In non-immune pregnant mice, biotin crossed the placenta and accumulated in fetal tissues, whereas fetuses from biotin-injected immune mice had less fetal biotin accumulation (Fig. 2c). Immunofluorescence using streptavidin-Alexa-488 demonstrated significantly higher biotin accumulation in the labyrinths of placentas from the immune mice (Fig. 2c).

Overall, these experiments suggest that maternal immune responses against β3 impede maternal to fetal placental exchange function.

**NK cells mediate immunopathology of the placenta in FNAIT.** To investigate the underlying mechanisms of placental abnormalities, morphometric and immunopathological studies of placentas were conducted. Histomorphometric studies of placentas from non-immune mothers showed proper decidual development at E14.5 with a decidua/labyrinth ratio ~0.4 (Fig. 3a). Decidua in placentas from immune mice was twice as large as in controls, significantly increasing the decidua/labyrinth ratio to 0.7. Decidual thickening, abnormal NK cell numbers, and reduced trophoblast migration were reported in prior work by other group[52]. In addition to their significantly greater abundance, uNK cells in immune mice showed higher perforin and NKp46 expression (Fig. 3b). More cell-free, perforin-reactive granules were observed in immune deciduas. In addition, percentage of activated uNK cells expressing markers of degranulation, such as CD107 (DBA⁺NKp46⁺CD107⁺ cells), was significantly increased in immune mice (Fig. 3c). To address whether NK cell accumulation and activation impair key processes of placentation in late gestation, we started anti-NK cell treatment at E11.5. Interestingly, we found that inhibition of activating receptor NKp46 or anti-Asialo-GM-1 depletion of NK cells significantly prevented miscarriage in immune mice at E14.5, allowing pregnancies to progress to term (Table 2). These observations suggest a pivotal role for activation of NK cells in FNAIT placental pathogenesis.

**β3 integrin-positive trophoblasts interact with NK cells.** First, we addressed expression of β3 integrin by human first-trimester placenta, first-trimester trophoblast cell lines (HTR-8/SVneo and SWAN 71), and by mouse placentas. Placenta sections from first-trimester human placentas (week 4.5, 5, and 9) expressed β3 integrin on syncytiotrophoblast and more importantly on invasive EVT as shown by double staining with HLA-G and cytokeratin-7 (Fig. 4a). Human umbilical vein endothelial cells (positive control) expressed β3 integrin, whereas platelet extract from β3⁻/⁻ mice did not (negative control; Fig. 4b). Human trophoblast cell lines expressed β3 integrin as did mouse placentas across gestation (E9.5–E18.5; Fig. 4b). Microscopic analyses of E14.5 implantation sites identified numerous *Dolichos biflorus* agglutinin (DBA)⁺ NK cells that appeared to interact with trophoblast cells expressing β3 integrin near the junctional zone. Significantly more interactions were present at the immune vs. non-immune maternal–fetal interface where uNK cells were less frequent and more senescent (Fig. 4c). NK cells accumulating at the feto–maternal interface in both non-immune and immune conditions were not proliferating cells (Fig. 4d). Most proliferating cells were located in the labyrinth, they appeared to be endothelial or trophoblast cells and their numbers were significantly decreased in immune vs. non-immune placentas (Fig. 4d).

NK cells resident in the decidua express both α1β1 and α2β1 integrins, which promote extracellular matrix attachment. To test the origin of DBA-positive cells accumulating in decidua in non-immune and immune states, α1 integrin staining was performed. uNK cells from E14.5 non-immune mice expressed basal levels of α1 integrin, suggesting a recent uNK infiltration (Fig. 4e), which is also supported by adaptive transfer of pNK cells (Supplementary Fig. 3). In contrast, uNK cells from immune mice expressed significantly higher levels (>2.5-fold) of α1, integrin suggesting that they are activated resident cells. In typical mouse pregnancies, most uNK cells die by senescence between E10.5 and E12.5 as observed in non-immune mice placentas (Fig. 4b). Anti-asialo-GM-1 treatment significantly decreased NK cell

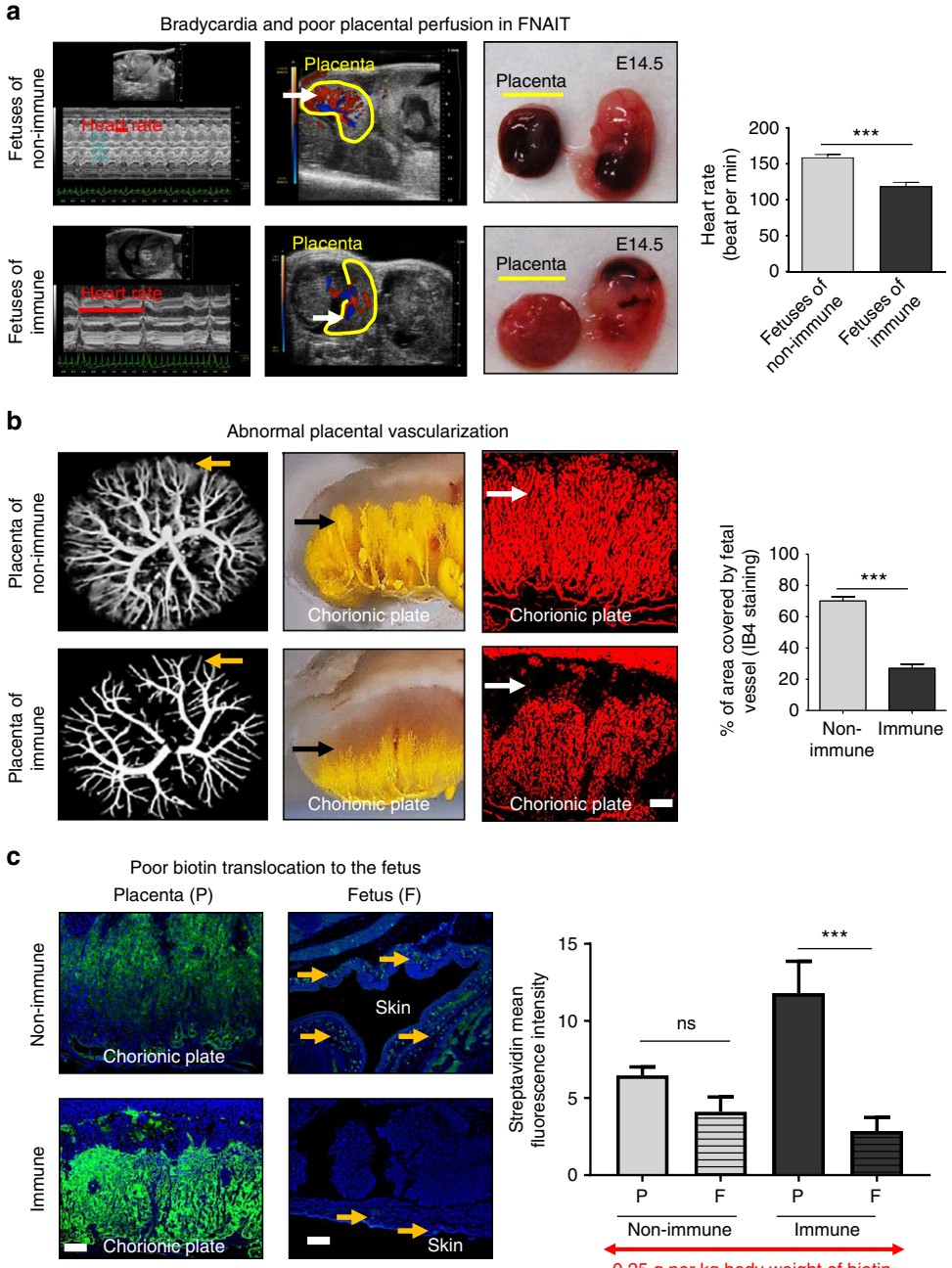

**Fig. 2** Abnormal placental vascularization and poor placental perfusion. **a** Bradycardia, impaired intraplacental blood flow (*white arrow* for Doppler waveforms) and poor placental blood perfusion (*gross pictures*) were found in FNAIT fetuses compared to fetuses of non-immune mice at E14.5. **b** Placental casts of the umbilical arterial circulation showed poor development of fetal capillaries in the labyrinth of placentas of immune mice as revealed by microtomography scan, stereomicroscopy micrograph, and IB4 immunostaining. **c** Fifteen minutes after maternal intravenous biotin injection, biotin transportation across the placenta into the fetal tissues (indicated by *yellow arrows*) was much more limited in fetuses from immune than control mice. Data were collected from more than eight pregnancies per group. Unpaired Student's *t*-test. Mean ± SEM. ***$p < 0.001$ and *ns* not significant. *Scale bars*: 200 μm (**b**, **c**)

accumulation in immune decidua but did not diminish α1 integrin expression by stromal cells.

**uNK cells induce trophoblast apoptosis in FNAIT.** To further investigate possible cellular and molecular mechanisms leading to placental insufficiency, SA remodeling was investigated. Significantly shorter SA diameters were documented in immune placentas (Fig. 5a). In E14.5 non-immune placentas, endovascular trophoblasts were found in significantly higher number in SA

(Fig. 5a). uNK cells in E14.5 immune placentas surrounded the SA and trophoblast migration into the vessels appeared to be impaired as trophoblast numbers were significantly decreased in the SA (Fig. 5a). Indeed, endovascular trophoblasts were scarcely present and smooth muscle actin (SMA) staining was markedly greater in the SA of immune placentas (Fig. 5a). SMA staining in non-immune placentas was low as expected in normal mouse pregnancies (mean fluorescence intensity = 6 ± 3). SMA staining was significantly higher in immune placentas (42 ± 11), but was normalized in placentas from asialo-GM-1-treated mice (7 ± 6).

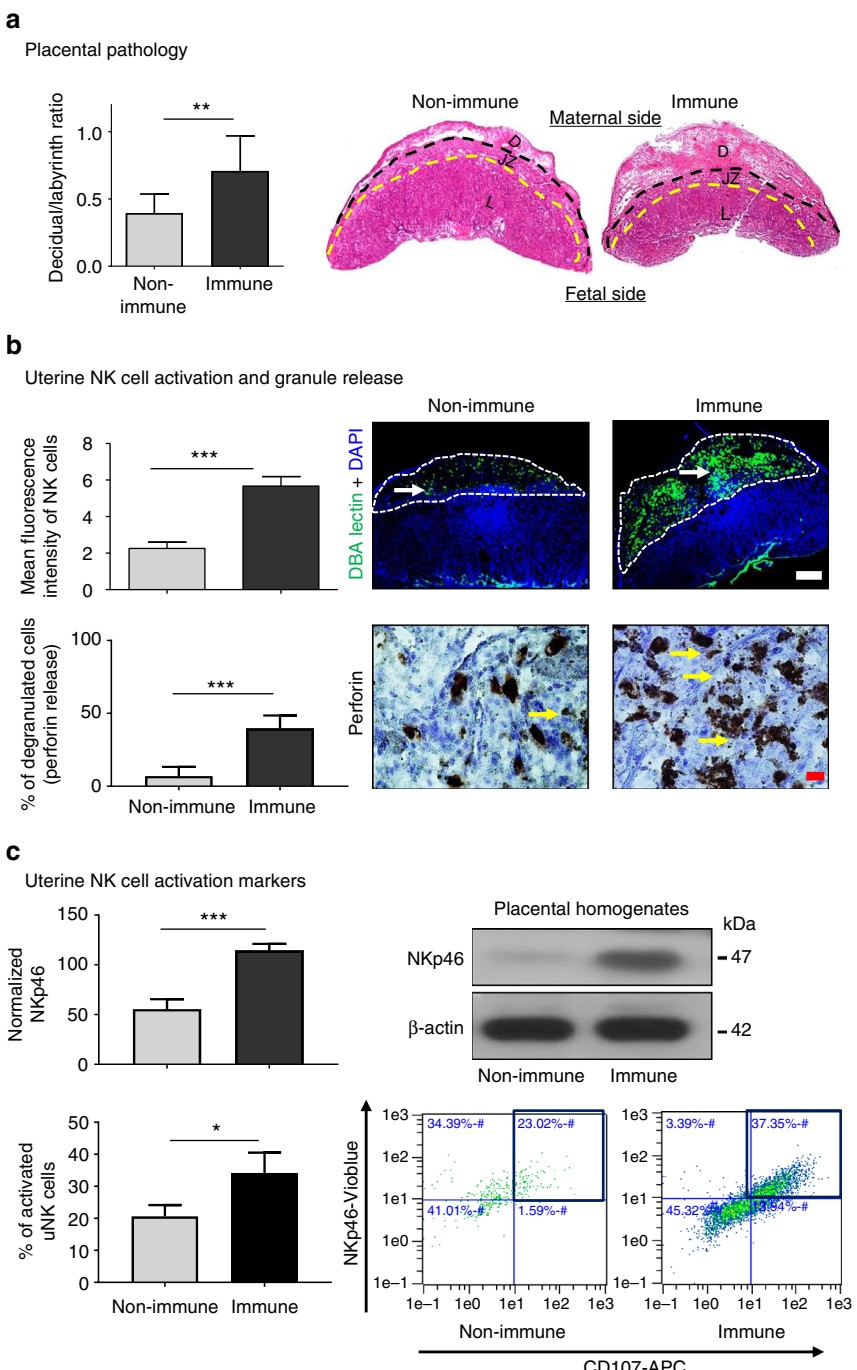

**Fig. 3** Placental pathology and NK cell accumulation in the decidua. **a** At E14.5, implementation sites from immune mice exhibited significantly enlarged decidua compared to non-immune mice placentas (*top panel*). The decidua/labyrinth ratio was significantly higher in placentas of immune mice. **b** DBA[+] NK cell number remained significantly elevated at E14.5 (*right top, immunofluorescence panel*) and their perforin granules were released (*right bottom, histology panel*, degranulated NK cells, *yellow arrow*). **c** NKp46 expression was significantly upregulated in placentas from immune mice (*top panel*). Percentage of activated uNK cells expressing markers of degranulation such as CD107 (DBA[+]NKp46[+]CD107[+] cells) are significantly increased in immune mice (*bottom panel*). Data were collected from more than eight pregnancies per group. *D* decidua, *JZ* junctional zone, *L* labyrinth. Unpaired Student's *t*-test. Mean ± SEM. *$p < 0.05$, **$p < 0.01$ and ***$p < 0.001$. *Scale bars*: (**b**) 500 µm *white color* and 20 µm *red color*

To address the basis for reduced endovascular trophoblast in the immune mice, dUTP nick-end labeling (TUNEL) stain was conducted. Significantly higher apoptosis of trophoblast was found in the spiral arteries of immune placentas (Fig. 5a). Anti-asialo-GM-1 treatment restored endovascular trophoblast survival in the SA, ultimately allowing pregnancy progression. In addition, western blots revealed that anti-asialo-GM-1 treatment reduced NKp46 expression in uNK cell lysates (Fig. 5b). To address whether NK cells can induce ADCC on EVTs, HTR-8/SVneo cells (5-chloromethylfluorescein diacetate, CMFDA-labeled in green) were co-cultured with uNKs isolated from placentas of immune mice in the presence or absence of anti-β3 IgG that cross-react human β3 integrin[48, 53] (Fig. 5b). NK cells from deciduas of immune mice induced greater apoptosis in these human trophoblasts (i.e., caspase-3 fluorescence) in the presence of anti-β3 IgG compared to control IgG, which was significantly

**Table 2 Inhibition or depletion of uNK cell prevents fetal death at late gestation**

| | Fetal survival (%) and 95% confidence interval | | | |
|---|---|---|---|---|
| **Treatment** | **E12.5** | **E13.5** | **E14.5** | **E15.5** |
| Immune + control serum | 34/34 **(100%)** <br> 0 | 31/34 **(91%)** <br> 10 | 18/35 **(51%)** <br> 17 | 4/34 **(12%)** <br> 11 |
| Immune + anti-NKp46 | 31/31 **(100%)** <br> 0 | 26/26 **(100%)** <br> 0 | 30/33 **(91%)** <br> 10 | 26/31 **(84%)** <br> 13 |
| Immune + anti-asialo-GM-1 serum | 32/32 **(100%)** <br> 0 | 28/28 **(100%)** <br> 0 | 31/31 **(100%)** <br> 0 | 38/40 **(95%)** <br> 7 |

Activating receptor blockade by anti-NKp46 antibody or NK cell depletion by anti-asialo-GM-1 serum prevents fetal death at E14.5, allowing the pregnancy to progress. Bold values represent survival percentage

alleviated by anti-Fcγ receptor blocker (2.4G2). uNKs from non-immune pregnant mice did not induce significant ADCC on similar target cells (Fig. 5b).

**Depletion or inhibition of uNK cells prevents miscarriage.** The question of whether targeted removal or disarming of NK cells could be a therapeutic strategy for prevention of FNAIT-induced miscarriage was addressed. In immune females bred by WT males where ~50% fetal loss occurs at E14.5, anti-asialo-GM-1 treatment starting from E11.5 resulted in birth of normally sized litters at term (n = 8; Fig. 6a). Of critical importance, anti-asialo-GM-1 antibody treatment by preventing placental abnormalities restored normal platelet counts in neonates and prevented severe bleeding often observed in neonates from untreated immune mice (Fig. 6a). To test whether maternal anti-β3 integrin antibody and NK cells result in ADCC in vivo, FcγRIIIas were blocked with anti-FcγRIIIa, 2.4G2 antibody. This treatment also prevented miscarriage (n = 5), albeit some fetuses developed ICH (not shown) and had low platelet counts (Fig. 6a). Pregnancies were also rescued following treatment with intravenous IgG (IVIG). Inhibition of the NK cell activating receptor NKp46 that contributes to perforin release[46] also significantly increased litter sizes at term in both severe (Fig. 6a) and moderate FNAIT (Fig. 6b). Passive transfer by intravenous injections at E10.5 of high titer anti-β3 IgG into non-immune β3−/− females bred with WT did not significantly induce miscarriage but the pups displayed bleeding disorders including ICH. These better outcomes of pregnancy suggest that manipulations of NK cells could be important in the management of FNAIT and other antibody-related pregnancy losses as already suggested by earlier studies[54].

**Discussion**

Bleeding disorders in fetuses and neonates, particularly ICH, have captured all of the clinical management attention in FNAIT[25]. There is a lack of knowledge regarding placental pathology and whether it provides the etiology for IUGR and fetal death[55, 56]. Here we establish the importance of IUGR/miscarriages in FNAIT. Strong evidence is provided that maternal alloantibody-associated placental pathologies dictate not only the severity of neonatal bleeding but importantly also the severity of IUGR/fetal death. Pathogenesis is based upon interactions between FcγRIIIa of uNK cells with anti-β3 integrin immune complexes on invasive trophoblasts that initiate ADCC. Both IUGR and miscarriage in our mouse model of severe FNAIT could be prevented by anti-asialo-GM-1-mediated NK cell depletion or by functional inhibition of the NK cell receptors FcγRIIIa or NKp46.

Although IUGR and fetal demise occur as frequently as ICH[5, 11], in human FNAIT, only chronic villitis has been reported as an FNAIT-linked placental anomaly. Chronic deciduitis is a serious inflammatory condition during pregnancy that is often associated with preterm labor[57]. In our murine model of FNAIT, we observed

fetal distress prior to death by ultrasound examination that involved placental (Doppler flow) and fetal (heart rate) compartments. We also documented placental inflammation (decidual enlargement, cytokine profiles, and NK cell hypercellularity with aberrant activation and prolonged survival in decidua basalis). Chronic deciduitis often results from fetal antigen stimulation causing placental inflammation. This aligns with our finding that paternally derived β3 integrin antigen is required for induction of placental inflammation and FNAIT in our mouse model. Th17 cytokines and deposition of maternal anti-β3 antibodies into decidua may play roles in recruitment of post mitotic NK cells from the peripheral circulation into decidua. In addition to inflammation in the maternal part of the placenta, we found that endothelial cells were reduced in the placental labyrinth, which showed vessel rarefaction and capillary deficiencies and that, trophoblast proliferation was significantly reduced in immune mice. Together, these features were shown to limit molecular transport from the maternal to the fetal compartment. This alone would be sufficient to explain fetal stress (bradycardia) and IUGR as reported previously[22]. These features, however, did not occur in isolation but were accompanied by apoptosis of invasive trophoblast, loss of angiogenic molecules, gain in anti-angiogenic signals, and failure of late gestational SA transformation. Each of these would additionally reduce maternal support of placental function and incrementally reduce fetoplacental nutrient availability. Whether or not the inadequate development of the fetal labyrinthine vasculature results from nutrient insufficiency will require further study.

uNK cells become locally "educated and licensed" at implantation sites and typically are committed to protect against placental infection while preventing fetal loss[58]. It is still questionable what triggers quiescent NK cells to become aggressive cytotoxic killers cells[46, 59, 60]. A fetus resulting from allogeneic mating expresses a variety of antigens that may serve as targets for rejection by the maternal immune system, yet inhibition of killer cells by hormones, stroma, and trophoblast cells typically prevents that from occurring[61]. Placental inflammation and immune complexes can alter this process considerably. The combination of uNK cells and maternal anti-β3 antibodies had a profound effect on survival of invasive trophoblasts and placental development. The ability of uNK cells to initiate and sustain placental inflammation and to effect ADCC against trophoblast is a dramatic paradigm shift in our understanding of pregnancy failure. For a long time, NK cells have been thought to be key contributors of placental vascularization including initiation of SA remodeling[62–64]. Tightly regulated interactions between NK cells and trophoblasts dictate normal SA remodeling as dNK cells prevent excessive invasion of trophoblasts and trophoblasts inhibit the maternal immune response and fetal rejection[29, 65]. It has been reported that FcγR is essential during uNK activation at mid-gestation in mice[41, 47, 66]. Given that maternal anti-β3 antibodies are capable of binding to trophoblast cells, the resulting immune complexes could be recognized by FcγRIIIa on uNK cells. This initiated a

cascade of outside-in signaling and uNK cell activation, and, ultimately, NKp46 and CD107 upregulation and perforin release. Subsequently, activated uNKs surrounded spiral arteries, induced apoptosis of invasive trophoblasts, and stunted SA enlargement.

The NK cells that accumulated in immune deciduas were not proliferating cells since they did not stain for Ki67. Given that uNK cell numbers typically decline after mid-gestation (E11.5),

we speculate that newly recruited killer cells come from peripheral blood. This is supported by our observation that expression of α1 integrin, which allows uNK cells to attach to the extracellular matrix, is increased three-fold in placentas of immune mice. Anti-asialo-GM-1-mediated NK cell depletion significantly reduced uNK cell number in the deciduas of immune mice. The remaining uNK cells in anti-asialo-GM-1-treated placentas did

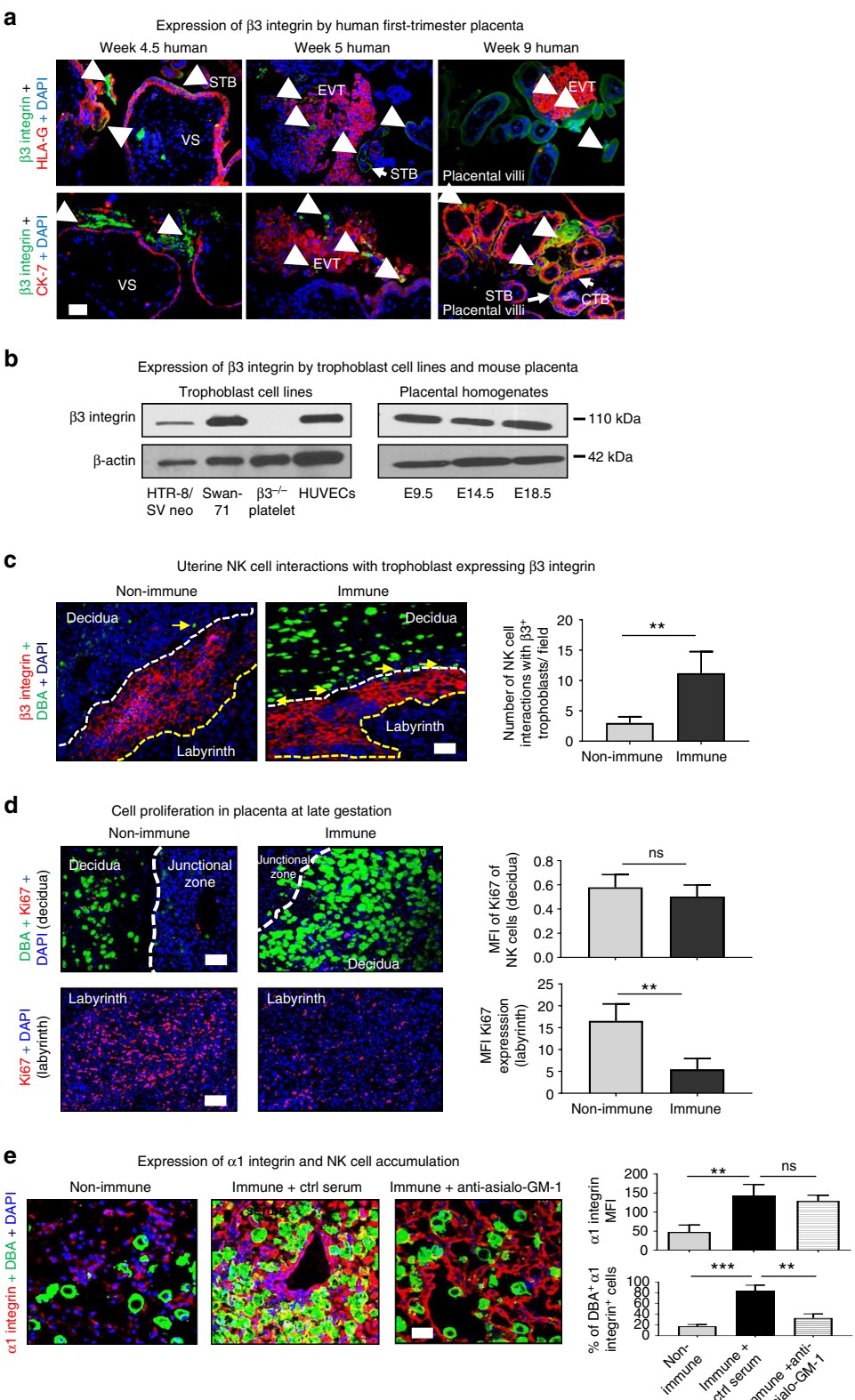

not cause placental pathology or fetal death, and both SA remodeling and trophoblast survival were restored to normal. More detailed, future phenotyping of the atypical, large, activated uNK cells seen at E14.5 should identify the origins and potential mixed composition of this pathogenic innate lymphoid cell population.

This report identified uNKs as cellular targets and their activating receptors FcγRIIIa and Nkp46 as molecular targets for potential therapeutic intervention in clinical FNAIT, and potentially other causes of antibody-related pregnancy loss. Anti-asialo-GM-1-mediated NK cell depletion is a proof of concept that aberrant maternal immune responses in the decidua can cause placental dysfunction. It is possible that macrophages cooperate with uNKs in inducing ADCC on trophoblast. In this respect the beneficial effect of anti-FcγRIII IgG and IVIG in preventing miscarriage could be achieved by blocking different Fcγ receptors on uNK cells, decidual macrophages, and dendritic cells thereby preventing placental inflammation and ADCC as previously reported[6, 10, 48]. Since IVIG is a costly drug with an unclear mechanism of action, its replacement by other, lower cost, and higher specificity drugs is appropriate. Anti-FcγRIIIa and anti-Nkp46 monoclonal antibodies had good efficacy in rescuing miscarriages in our mouse model. Whether these treatments could be useful in humans warrants further investigation. In summary, maternal immune responses to fetal platelet antigens cause IUGR and miscarriage linked with vascular and immune pathologies in placentas. Targeting of NK cells in FNAIT that rescued miscarriage and reduced severity of bleeding in fetuses/neonates should have great translational importance.

## Methods

**Mice**. β3$^{-/-}$ mice were kindly provided by Dr. Richard O. Hynes (Massachusetts Institute of Technology, Boston, MA, USA). All mice were housed in specific pathogen-free research vivarium at St. Michael's Hospital, Keenan Research Centre after protocol approval by the St. Michael's Hospital Animal Care Committee. Permission was granted to perform animal experiments by the animal care committee). All procedures were in compliance with Guidelines of the Canadian Council on Animal Care. β3$^{-/-}$ mice were backcrossed to a BALB/c background 10 times and bred to generate syngeneic gene-deficient mice. Experiments were performed when female and male mice were 7–10 weeks of age. BALB/c WT mice (stock number 028) were purchased from Charles River Canada.

Our FNAIT model was previously described[6, 7, 10, 48, 67]. Briefly, β3$^{-/-}$ females were immunized twice, 1 week apart via tail vein injections of gel-filtered WT platelets ($1 \times 10^8$ for severe FNAIT or $1 \times 10^7$ for moderate FNAIT). Immunized β3$^{-/-}$ females were bred with WT males (FNAIT group here after referred to as immune). The control group of β3$^{-/-}$ females was non-immune and mated by WT males. Immune and non-immune pregnant mice, their implantation sites, and neonates were compared at matched time points. Gestation E0.5 corresponds to the day a copulation plug was found. Immune females mated by β3$^{-/-}$ males were used as a FNAIT-negative control to demonstrate that β3 expression in placenta is required for induction of miscarriage. Immune females mated by β3$^{-/+}$ males were used to demonstrate that ADCC occurred only in β3$^{-/+}$ but not in β3$^{-/-}$ conceptuses from the same mother. Additionally, injections of anti-β3 immune sera into non-immune mice (bred by WT male) served as a passive FNAIT model to test whether only anti-β3 can induce or not ADCC.

**Reagents**. Rabbit anti-mouse perforin (Cat# 3693, dilution ($d$) = 100), active (cleaved) caspase-3 (Cat# 9664L, clone: 5A1E, $d$ = 100), β1 (Cat# 34971S, clone: D6S1W, $d$ = 100, and β3 integrin (Cat# 13166S, clone: D7X3P, $d$ = 100) antibodies were obtained from Cell Signaling Technology (Pickering, ON, Canada). Alexa Fluor 488-conjugated goat anti-rabbit IgG (Cat# A-11034, $d$ = 200), Alexa Fluor 594-conjugated isolectin GS-IB4 (Cat# I-21413, $d$ = 200), and Alexa Fluor 594-conjugated goat anti-rabbit IgG (Cat# R-37117, $d$ = 200) were from Invitrogen Canada (Burlington, ON, Canada). Biotinylated DBA (Cat# B-1035) was from Vector Laboratories (Burlington, ON, Canada), avidin-Alexa-488 (Cat# A-21370, $d$ = 200), and streptavidin-Cy3 (Cat# 438315, $d$ = 200) were from Thermo Fisher (Mississauga, ON, Canada). The angiogenesis array (Cat# ARY015) was purchased from R&D Systems (Burlington, ON, Canada). Anti-CD107-APC (Cat# 130-102-191, clone: 1D4B, $d$ = 100), anti-NKp46-vioblue (Cat# 130-103-137, clone: 29A1.4.9, $d$ = 100), anti-CD4-APC (Cat# 130-109-494, clone: REA604, $d$ = 100), anti-CD3e-APC (Cat# 130-102-791, clone: 145-2C11, $d$ = 100), and anti-IL17-PE (Cat# 130-103-015, clone: TC11-18H10, $d$ = 100) were obtained from Miltenyi Biotec (Auburn, CA, USA). For placental casts, a radio-opaque silicone rubber X-ray contrast agent (Microfil, Cat# MV-122) was obtained from Flow Tech (Carver, MA, USA). Anti-mouse NKp46 was from Biolegend (Cat# 137614) and rabbit anti-asialo-GM-1 (Cat# CL8955) was from Cederlane (Burlington, ON, Canada); anti-FcγRIIIa (2.4G2; Cat# 553140), non-immune rabbit IgG (Cat# 50875), and non-immune rat IgG (Cat# 553993) were from BD Bioscience (Mississauga, ON, Canada). Rabbit α1 integrin antibody (Cat# ab181434, $d$ = 100), SMA (Cat# ab5694, $d$ = 100), and pan-cytokeratin (Cat# ab20206, $d$ = 100) were from Abcam (Cambridge, MA USA). In Situ Cell Death Detection Kit (Cat# 11684795910) was from Roche Diagnostics (Canada, Laval, QC, Canada). Trophoblast cell lines (HTR-8/SVneo representing first-trimester human EVTs[60] and Swan 71, representing telomerase-induced immortalized first-trimester human trophoblasts) were from Dr. Stephen Lye lab, Lunenfeld-Tanenbaum Research Institute, Mount Sinai Hospital, and University of Toronto. These were maintained between passage 14 and 20, and they retained hallmarks of first-trimester trophoblast. HTR-8/SVneo was cultured in RPMI-1640 medium, whereas Swan 71 in DMEM F-12K. These media were supplemented with 10% fetal bovine serum (FBS), 100 IU ml$^{-1}$ of penicillin, and 100 μg ml$^{-1}$ of streptomycin (Invitrogen) and cells were incubated at 37 °C with 5% $CO_2$.

**Ultrasound imaging followed by biotin injection**. To identify the gestational age at which fetal loss occurred, serial high-frequency ultrasound imaging of each pregnant mouse to assess fetal vitality, cardiac function, and intraplacental blood flow was performed at embryo days E.11.5, E.13.5, E14.5, and E.15.5 as previously described[7]. Since ultrasound revealed that fetal deaths occurred at E14.5 ± 1.5, all subsequent experiments were carried out at E14.5. Pregnant mice were anesthetized with 2% isoflurane (inhaled) and maintained on 1% during imaging. After abdominal shaving, chemical depilation (Nair™, Church & Dwight Co, Inc.) and application of coupling gel (37 °C), high-frequency ultrasound imaging was performed by nonlinear contrast imaging using a Vevo® 2100 system and a MS550 transducer (32–56 MHz). After ultrasound, some animals were intravenously injected with biotin (0.25 g per kg body weight) to test maternal–fetal transport of essential nutrient[51]. Biotin is transported by its sodium-dependent multivitamin transporter[68]. In normal placenta intravenous biotin injection to the mother can be detected in the fetuses after 15 min. Animals were killed by cervical dislocation 15 min later. Placentas and fetuses were harvested from these mice and processed for biotin staining with streptavidin.

**Contrasted micro-CT scanning**. Microfil was infused into the fetoplacental vasculature as previously described[52]. Briefly, individual E14.5 implantation sites were removed from chilled uteri. Fetus and placenta were surgically exposed and bathed in 37 °C phosphate-buffered saline (PBS) to resuscitate the fetus and resume circulatory function. A double-lumen glass cannula was advanced into the umbilical artery. Blood

**Fig. 4** β3 integrin expression in early human placenta. **a** β3 integrin was detected on multinucleated syncytiotrophoblast cells (STB, cytokeratin-7$^+$) and on extravillous trophoblast cells (EVT, HLA-G$^+$) were found in normal human placental villi from the first trimester of pregnancy (4.5–9 weeks). **b** Western blotting showing expression of β3 integrin by trophoblast cell line (*right panel*; HTR-8/SVneo and Swan 71) as well as mouse placenta (*left panel*). **c** At the maternal–fetal interface, fetal allogeneic trophoblast cells interact with maternal killer cells in the decidua. Trophoblasts (glycogen cells) in the junctional zone also showed more interactions with NK cells (*yellow arrows*) in immune mice placenta (*left panel*). These interactions were rare in the decidua of non-immune mice where the NK cells near junctional zone appeared senescent (*right panel*). **d** uNK cells from both non-immune and immune mice were not proliferating cells (*top panel*). The proliferating cells (Ki67-positive cells) were localized in the labyrinth and their numbers were significantly reduced in placentas of immune compared to non-immune mice (*bottom panel*). **e** Non-immune mice had lower levels of α1 integrin expression (*top panel, left*). Placentas from immune mice showed significantly upregulated α1 integrin expression, suggesting stronger tethering of uNK cells to extracellular matrix (*top panel, center*). Anti-asialo-GM-1 treatment reduced uNK cell accumulation (*top panel, right*). Data were collected from eight pregnancies per group. HLA-G marks only extravillous trophoblast, whereas cytokeratin-7, an epithelial cell marker, reacts with both STB and EVT. *CTB* cytotrophoblast, *VS* villous stroma. Unpaired Student's *t*-test. (**a**–**d**); two-way ANOVA followed by Bonferroni post hoc test (**e**). Mean ± SEM. **$p < 0.01$, ***$p < 0.001$, *ns* not significant. *Scale bars*: 100 μm (**a**, **e**) and 50 μm (**c**, **d**).

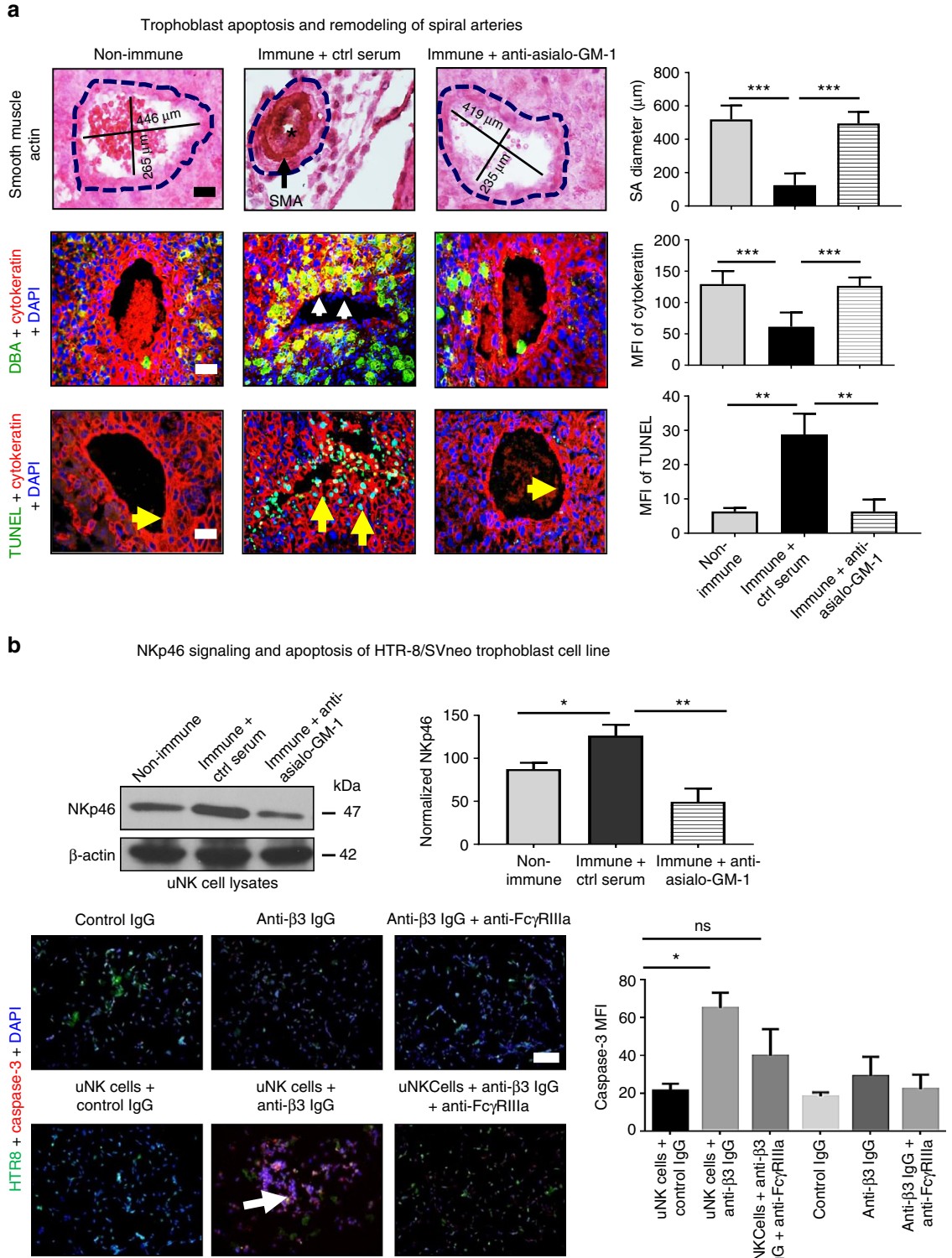

**Fig. 5** Mechanisms of NK cell-mediated trophoblast apoptosis. **a** SA diameters were significantly larger in both non-immune and anti-asialo-GM-1-treated immune mouse placentas, whereas SA diameter was significantly reduced in placentas of non-treated immune mice, which showed higher expression of smooth muscle actin (*black arrow, top histology panel*). Spiral artery (SA) remodeling in placentas from non-immune mice showed presence of infiltrated endovascular trophoblasts. uNK cells surrounding maternal vessels impaired trophoblast migration into SA of immune mice. This was significantly ameliorated by anti-asialo-GM-1 treatment, which reduced NK cells' accumulation in the placenta (*middle immunofluorescence panel*). TUNEL staining of placentas from immune mice further confirmed apoptosis of trophoblasts surrounding SA. These abnormalities were significantly ameliorated by anti-asialo-GM-1 treatment (*bottom immunofluorescence panel*). **b** NKp46 signaling was significantly upregulated in uNK cell lysates from immune mice compared to non-immune. Anti-asialo-GM-1 treatment also reduced NKp46 expression (*top panel*). Purified uNK cells from immune pregnant mice induced antibody-mediated cell cytotoxicity on a cultured HTR-8-SV/neo trophoblast cell line, which was reduced by treatment with anti-FcγRIIIa antibody (2.4G2; *bottom panel*). In vivo data are representative results of five pregnancies per group at E14.5 (**a**, **b**), whereas in vitro data are four sets of experiments (**b**). Unpaired Student's t-test. (**a**); two-way ANOVA followed by Bonferroni post hoc test (**b**). Mean ± SEM. *$p < 0.05$, **$p < 0.01$, ***$p < 0.001$, *ns* not significant. *Scale bars*: 50 μm (**a**) and 200 μm (**b**)

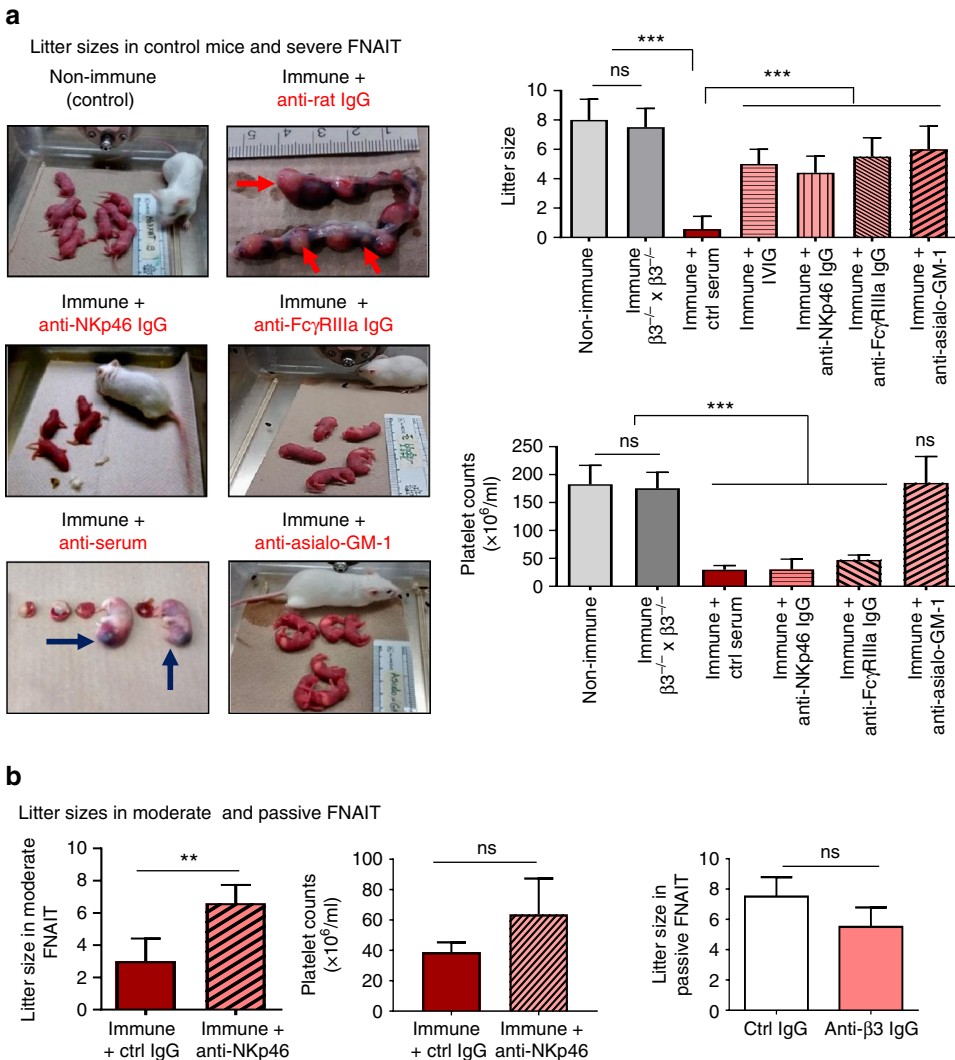

**Fig. 6** Inhibition of activating receptors and NK cell depletion prevent IUGR/fetal death. **a** Non-immune mice had normal pregnancies and delivered healthy pups. Litter sizes were not affected in immune females mated by $\beta3^{-/-}$ males. In immune mice, miscarriages (*red arrows*, resorbed fetuses) were frequent as well as severe bleeding in neonates (*blue arrows* for ICH) and low neonatal platelet counts. NK cell depletion induced by anti-asialo-GM-1 serum prevented miscarriages and bleeding in neonates. Inhibition of NK cell activation through NKp46 or FcγRIIIa blockade or IVIG ameliorated pregnancy outcomes (litter size). **b** Anti-NKp46 treatment in moderate FNAIT (induced by lower dose of $10^7$ platelet transfused/immunization) also prevented miscarriages. Injections of high titer anti-β3 integrin IgG sera into non-immune mice did not cause miscarriage, albeit some neonates developed bleeding and ICH. Data were collected in four to eight pregnancies per group. Unpaired Student's *t*-test (**b**); two-way ANOVA followed by Bonferroni post hoc test (**a**). Mean ± SEM. **$p < 0.01$, ***$p < 0.001$, *ns* not significant

was cleared from the vasculature using heparinized saline with xylocaine. The contrast agent was perfused into the arterial vasculature until its bright yellow color could be seen in the capillary bed, and then the vessels were tied off to maintain pressure. The silicone rubber was allotted time to polymerize before the umbilical cord was severed and the placenta was immersion-fixed (10% buffered formalin phosphate; 48 h; 4 °C). Specimens were mounted in 1% agar made with 10% formalin for scanning. Three-dimensional (3D) data sets were acquired for each specimen using an MS-9 micro-CT scanner (GE Medical Systems, London, ON, Canada).

**Histology and TUNEL staining.** For histological analysis, placental tissue was fixed in formalin, paraffin-embedded, sectioned (8 µm), transferred to glass slides, and stained with hematoxylin and eosin using standard techniques. For immunohistochemistry staining and immunofluorescent, paraffin-embedded tissue sections of mouse placenta and human placenta samples from Toronto Biobank were process for antigen retrieval. uNK cells and HTR-8/SVneo cells that had been co-cultured on coverslips were fixed with paraformaldehyde (PFA, 4% w/v, 5 min). Placenta sections and cultured cells were blocked in 5% bovine serum albumin and then incubated with the primary antibody of interest overnight. After washing, the secondary antibody (anti-rabbit IgG-Cy3 or Alexa Fluor 594-conjugated goat anti-rabbit IgG) and avidin alexa-488 or avidin-Cy3 conjugated were incubated for 1 h, as previously described[7]. Apoptosis was detected in tissue sections via TUNEL

staining with an *In Situ* Cell Death Detection Kit, according to the manufacturer's protocol. Briefly, paraffin-embedded placental sections were fixed in PFA (4%, 5 min, room temperature (RT)) and permeabilized. Positive controls were treated with 100 U DNase I for 30 min at 37 °C. Slides were incubated with reaction buffer for 1 h, wash three times in 0.1% PBS-T, and then mounted in Vectashield mounting medium containing 4′,6-diamidino-2-phenylindole dihydrochloride (DAPI) for nuclear counterstaining. Fluorescent images were acquired using Olympus upright or Zeiss confocal microscopes and analyzed by Image J software (US National Institutes of Health, Bethesda, MD, USA).

**In vivo anti-NK cell treatments and passive FNAIT models.** NK cell depletion or inhibition of the NK cell activation receptors (NKp46 or FcγRIIIa) started at E11.5. Intraperitoneal injection of 50 µl of undiluted anti-asialo-GM-1 serum was used for NK cell depletion at E11.5 and E14.5 (*n* = 8). Anti-NKp46 (3.33 mg per kg body weight) and anti-FcγRIIIa (6.66 mg per kg body weight)[69] treatments were given intravenously at E11.5 and E14.5 (*n* = 4). IVIG, a commonly used drug for FNAIT management, was compared to anti-NK therapies. IVIG (1 g per kg body weight)[6] was intraperitoneally injected into immune mice twice at E.0.5 and E7.5 (*n* = 6).

**NK cell co-culture with trophoblast cell lines.** Implantation sites were collected and kept over ice in small plate containing RPMI medium. After microscope-aided

dissection, decidua and mesometrial lymphoid aggregate of pregnancy were dissected, each tissue type was pooled by litter and finely minced. Tissue was then transferred into Hank's solution (37 °C) containing DNase I for enzymatic dissociation. Every 10 min, the solution was gently pipetted up and down with Pasteur pipettes. After 1 h incubation time, the slurry was passed through a nylon tissue strainer to obtain a single-cell suspension, and then washed twice in cold medium, and enumerated for viability and cell number. uNK cells were prepared by DBA-positive selection as previously reported[70]. Briefly, red blood cells were lysed by 3.5% NaCl. Decidual and stroma cells were removed by adherence after centrifugation (400 × g, 5 min). Placental leucocytes were recovered and then separated by density gradient centrifugation (Lympholyte®-M; 1400 × g; 30 min). The leucocyte preparation was washed in 2% FBS in 1× PBS and loaded on DBA-lectin-coated magnetic beads for 15 min at 4 °C. Retained uNK cells were eluted after a 5 min incubation with 2% FBS in 1× PBS containing 0.1 M N-acetyl-D-galactosamine and pelleted in RPMI-1640 supplemented with 10% FBS and 1 mg ml$^{-1}$ gentamycin for culture. To assess cytotoxicity, NK cells were co-cultured with cell tracer CMFDA-stained HTR-8/SVneo cells at a 2:1 ratio. After 24 h incubation, anti-caspase-3 was utilized to detect cell death. Peripheral NK cells were purified by positive selection using anti-NKp46 antibody and cells were labeled with cell tracer fluorescence dye (5-(and-6)-Carboxyfluorescein Diacetate, Succinimidyl Ester), then injected to pregnant mice at E13.5 and tracked within the recipient organs.

**Detection of T helper-17 and uNK cells by cytometry**. Spleens from pregnant mice at E14.5 were enzymatically digested and passed through a nylon tissue strainer to obtain a single-cell suspension. Red blood cells were lysed by 3.5% NaCl. Spleen leucocytes were obtained by density gradient centrifugation (Lympholyte®-M; 1400 × g; 30 min). Splenocytes or uNK cells were incubated for 30 min at 22 °C with the following antibodies: anti-CD4-APC, anti-CD3e-APC, and anti-IL17-PE (splenocytes) or DBA-Alexa-488, NKp46-Vioblue and CD107-APC (uNK). Cells were analyzed by flow cytometry (Miltenyi), at least 20,000 events were scored and analyzed with Flow Jo software. Single cells were gated for CD4 (Vioblue) and CD3e (APC) to select double-positive cells. Then, these double-positive cells were gated on PE for IL17. The percentage of CD4$^+$CD3e$^+$ IL17$^+$ cells corresponds to triple-positive cells from the total splenocyte preparations (Supplementary Fig. 4). Single cells were gated on Alexa-488 to select DBA$^+$ cells, and then these cells were gated on Vioblue and APC to select NKp46 and CD107 double-positive cells CD107.

**Western blotting and angiokines cytokine array analyses**. In preparation for western blotting, HTR-8/SVneo and Swan 71 human trophoblast cell lines were harvested from confluent flasks. These cells and mouse placenta tissue were lysed in radioimmuno-precipitation assay buffer, supernatant collected after centrifugation (×10,000 g) and stored at −20 °C until western blotting. Protein (25 µg) from the cell lysates or placenta homogenates was analyzed by western blotting, as previously described[7]. Briefly, protein samples were electrophoresed on 8% SDS polyacrylamide gels and electrotransferred onto polyvinylidene fluoride membranes. Immunodetections were carried out with specific antibodies of interest (Supplementary Fig. 5). For the angiogenic array, E14.5 mouse placenta homogenates were subjected to immunoblotting as recommended by the manufacture (R&D System). Immobilized antigens were detected by chemiluminescence using horseradish peroxidase-conjugated secondary antibodies, an ECL kit, and autoradiography on Kodak film.

**Statistical analysis**. Data shown are mean ± SEM. Statistical comparisons were made using an unpaired Student's t-test or two-way ANOVA followed by Bonferroni post hoc test as appropriate. Densitometry and immunofluorescence analysis was performed using Image J software (National Institutes of Health) and values were normalized to internal control or total β-actin. In comparing fetal survival 95% confidence interval for proportion were calculated using litter size and percentage of alive fetuses (https://www.allto.co.uk/tools/statistic-calculators/confidence-interval-for-proportions-calculator/). In each pregnancy, three to four placentas or fetuses from same mother were analyzed and averaged (n). Statistical analysis was performed on the mean values from several mothers (N). Differences were considered statistically significant when p < 0.05.

**Data availability**. The data that support the findings of this study are available from the first author and the corresponding author upon reasonable request.

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

## Acknowledgements

We gratefully acknowledge Dr. Richard O. Hynes for providing the β3 integrin-deficient mice. We thank Dr. John W. Semple for stimulating discussion during manuscript preparation. This work was supported by Canadian Institutes of Health Research (MOP 68986 and MOP 119551). Equipment was supported by funds from the Canada Foundation for Innovation, St. Michael's Hospital, and Canadian Blood Services. I.Y. is a recipient of a Canadian Blood Services postdoctoral fellowship.

## Author contributions

I.Y. designed and performed all the experiments, analyzed and interpreted data, and prepared and revised the manuscript. W.-S.T., D.Z., B.E.O., S.L. and G.Z. conducted some experiments and prepared the manuscript. H.L.-P., D.Q., L.Y., P.H., X.-Y.W. and D.J.S. provided guidance for study design and data interpretation. C.D., J.Z., J.G.S., S.J.L., J.B. and C.P. provided guidance, cell lines, and human placenta sections for the study and data interpretation. B.A.C., S.L.A. and J.F. contributed to study design and data interpretation, as well as manuscript preparation. As the principle investigator, H.N. supervised study design, data interpretation, and manuscript preparation.

## Additional information

**Competing interests:** The authors declare no competing financial interests.

