## [Peer Review File · Nature Communications]

Reviewers' comments:

Reviewer #1 (Remarks to the Author):

In this report, the authors investigate the role of natural killer (NK) cell signaling in fetal and neonatal alloimmune thrombocytopenia (FNAIT). The experimental approach utilizes an *Itgb3* null mouse model transfused with wild type platelets and then bred to wild type males. The wild type platelet transfusions stimulated an immune response in the *Itgb3* null females and the development of anti-beta 3 integrin antibodies, which resulted in IUGR and ultimately fetal death at ~E14.5. The experimental work is informative and the results are of considerable interest. Some concerns regarding the experimental design, presentation and interpretation of the data, including previously published reports, are presented below.

1. Abstract and elsewhere in the manuscript: Extravillous trophoblast is a term used to describe a lineage of trophoblast cells from the human placenta. The term is not appropriate for describing trophoblast lineages in the mouse placenta. EVT is used as an abbreviation of extravillous trophoblast and is used inappropriately to describe invasive trophoblast/endovascular trophoblast throughout the manuscript.
2. Page 4, paragraph 3, lines 1 and 2: Inappropriate reference for describing interactions between NK cells and trophoblast in the mouse and human. The reference only includes human data and the findings are based entirely on in vitro experimentation, which is of concern.
3. Page 7, lines 4-5: The "not shown" data on PGF and sFlt-1 should be presented in the Supplemental figures.
4. Page 8, paragraph 2: What was the rationale for using biotin transfusion to monitor placental exchange? Biotin is transported by the SMVT protein (Ganapathy et al. J Pharm Exp Therapeutics 294:413-420, 2000). Does the activity of the SMVT transporter reflect all placental transport function? The authors should provide references or describe the logic why monitoring biotin transport across the placenta is an informative method for monitoring placental exchange. Where is the western blotting for biotin transfer data presented?
5. Page 9, 2nd paragraph: What the rationale for using "first trimester" trophoblast cell lines rather than first trimester primary trophoblast cells? The validity and appropriateness of the Htr8 and Swan71 cell lines have been questioned as models for first trimester human trophoblast cells (see Lee et al. What is trophoblast? a combination of criteria define human first-trimester trophoblast. Stem Cell Reports 6:257-272, 2016). Furthermore expression profiles could be performed on first trimester human trophoblast tissues rather than relying entirely on cell lines.
6. Page 10, 3rd sentence: awkward - ... a fewer recent dNK infiltration....
7. Page 10, paragraph 2: The results presented seem to contradict the findings of Croy and co-workers, which indicate that NK cells promote uterine spiral artery remodeling, including disappearance of vascular smooth muscle. Consequently, is the conclusion that there are more NK cells, which persist but these NK cells are actually ineffective in their fundamental actions on uterine spiral artery remodeling and instead act primarily to prevent trophoblast-guided uterine spiral artery remodeling? Thus it would appear that the NK cells persisting in the uterus represent a unique subset of NK cells. Can this subset of NK cells be better defined based on surface marker expression?
8. Page 14, 2nd to last sentence: Ref Nos. 31, 53 are inappropriate for the statement. These reports

do not indicate that dNK cells prevent/restrain invasion of EVT. Instead these reports describe cooperative roles for dNK cells promoting/facilitating EVT invasion using in vitro models. The best in vivo experimental evidence for NK cells restraining EVT invasion was generated in the rat (Chakraborty et al. PNAS 108:16295-16300, 2011).

9. Page 17, paragraph one: Using wild type mice purchased from a vendor as a control for mice maintained as a colony is potentially problematic. Mouse colonies maintained at separate facilities show genetic drift overtime and thus introduce additional variables into the experimentation.

10. Page 18, paragraph one, last sentence: Some references for the Htr8 and Swan71 cell lines. Additionally how were the cells maintained? How many passages? Do the cells used for the experiments retain hallmarks of 1st trimester trophoblast?

11. Page 19, paragraph one, last two sentences: What is the logic for monitoring biotin transfer? More detail needs to be included for the procedure, especially monitoring biotin in the placenta and fetus. Inclusion of references describing the procedure would be helpful.

12. Page 20, paragraph two: How was the amount of antibodies used to neutralize NK cells? Is this based on earlier experimental work or were pilot experiments performed to optimize the antibody dosage? Some quantitative data should be included on the effectiveness of the antibodies in decreasing NK cell numbers.

13. Page 21, Why DBA negative selection to isolate uterine NK cells? Are the NK cells used in the experiment DBA positive or negative?

14. Fig2B: The y-axis for the bar graph is confusing.

15. Fig. 2C: What is presented in the bar graph? MFI is not defined.

16. Fig. 3B and elsewhere: MFI is not defined.

17. Fig. 4B: How were the junctional zone cells identified as glycogen cells? It would seem that the critical NK cell interactions will be with invasive trophoblast.

Reviewer #2 (Remarks to the Author):

This manuscript tests the hypothesis that the miscarriages and intrauterine growth restriction (IUGR) observed in fetal/neonatal alloimmune thrombocytopenia are due to an inflammatory response and placental pathology mediated by decidual NK (dNK) cells, through antibody dependent cell cytotoxicity (ADCC). The authors conclude that antibody production against allogeneic human platelet antigen (HPA)-1a, specifically in integrin $\beta 3$, starts an inflammatory response and mediates ADCC by dNK. Activated dNK are then responsible for trophoblast apoptosis, poor vascular remodeling, and pregnancy failure. The authors present clear in vivo data demonstrating that in the presence of anti-integrin $\beta 3$ antibodies, there is an increase in inflammation, reduction of growth factor production, IUGR and reduced fetal survival. Placental perfusion and vascularization, as well as nutrient transfer to the fetus, are also reduced. The authors also demonstrate an increase in NK cell numbers and

granularity, as well as interactions with trophoblast, and show that the phenotype can be reversed by NK cell depletion, blocking of the natural cytotoxicity receptor NKp46, or blocking of Fc receptors. The *in vivo* mouse data is convincing, but the manuscript has an important flaw – it is not easily translatable to humans. The authors do not show clearly if the effect is due to dNK or to infiltrated peripheral blood NK, which is important since dNK do not express Fc receptors. In addition, the organization of the figures and legends is confusing, and some of the methods are not clear.

1. Nomenclature – use uNK when mentioning mice, dNK is for humans. Also, EVT is a human term.
2. “These females lost their heterozygote fetuses around E14.5 and delivered litters of $\beta 3^{-/-}$ offspring that were half the size of litters of non immune mice”. How was this determined? Were the mice sacrificed at E14.5 and the resorptions phenotyped? Or was this just an assumption?
3. The figure legends are insufficient in most cases. The descriptions are not enough to understand what certain plots within panels of the same figure represent. As an example, in figure 3B, the right panel (western blot and bar plot) is not well described. What is “control”? How were this lysates obtained? It would be better to add more letters or to describe the panels (right, left, top bottom) in more detail.
4. The figure organization should also be improved. In many cases, such as figure 5, the panels do not follow the order in which they are described in the manuscript.
5. The main weakness of the manuscript is figure 5B. The authors test the activity of murine NK cells against human cells. This is a xenogeneic interaction. Any mouse antibody that binds a protein expressed in the human cell membrane could induce ADCC in the murine NK cell, since there is no MHC inhibition. This experiment should be scraped altogether or done using a mouse trophoblast cell line (e.g. SM9-1). In addition, the method used to measure cytotoxicity is caspase-3 fluorescence; a much better method would be a ^{51}Cr release assay.
6. The premise of this manuscript unfortunately does not translate to humans. The authors seek to prove that that the fetal/placental pathology observed in FNAIT is caused by ADCC mediated by uterine NK cells. Several of the experiments do suggest that the inflammation, loss of vascularization, overall placental pathology and fetal loss are NK cell – dependent (NK cell depletion and NKp46 blocking rescue the phenotype) and specifically due to the presence of anti- $\beta 3$ antibodies that lead to ADCC by NK cells (Fc blocking also rescues the phenotype). However, human decidual NK cells do not express Fc receptors. Thus, although the mouse phenotype is convincing, the human disease is probably not dependent on NK cell ADCC. Unless the authors assume that somehow peripheral blood NK cells infiltrate the placenta, but in that case the ADCC is not due to Fc receptors on dNK – it’s pNK-dependent. This is shown by the lack of ADCC by dNK of non-immune mice towards human trophoblast cell lines – these dNK are residents, not recently infiltrated pNK, and do not express Fc receptors. There is also the possibility that the antibody production induces an inflammatory response that leads to uterine NK cell activation and expression of Fc-receptors. In this case, NK cells do not initiate inflammation, but can still be responsible for the placental pathology.
7. To identify the origin of the increased NK cell population in “immune” mice, I suggest a procedure similar to the one used in Chiossone et al (2014) *Haematologica* 99(3) 448-457. Peripheral blood NK cells from EGFP+ mice were adoptively transferred to pregnant mice and tracked within the recipient mice. This way the authors can conclude whether the cytotoxic NK cells are activated dNK or infiltrated pNK that express Fc receptors.

Reviewer #3 (Remarks to the Author):

The paper by Yougbaré et al demonstrate in my opinion quite clearly that fetal / Neonatal alloimmune thrombocytopenia (FNAIT) involves decidual NK (cytotoxic) activation and persistence of these cells.

- 1) They demonstrate clearly that in a murine FNAIT model, the alloimmune thrombocytopenia is

correlated with placental and neonatal pathologies, including lower foeto placental weight (Intrauterine growth restriction, IUGR), enhanced abortion /resorbtion rates, placental and decidual bed vascular anomalies (improper development of spiral arteries), a Th-17 bias, anomalies of Sflt-1 and angiogenic cytokine production. This was confirmed by Doppler assessment of placental vascular flow , all that confirmed by (gross ! very gross) anatomical examination of the placenta, and functional studies of placental exchange functions by biotin transfer test. This pathology correlates with an INCREASED decidual bed, itself filled by activated decidual NK cells (dNK), with in case of pathology high NKP46 and perforin expression, indicating an activation towards the cytotoxic pathway rather than engagement in the activation of the trophic and angiogenic pathway. Inhibition of NKP46 or depletion of the excess NK cells by Asialo GM1 reduced this pathology.

2) The NK accumulation and persistence correlates with expression of $\alpha 1$ dNK integrin, nicely suggesting an infiltration process. However, anti Asialo GM1 antibody, while reducing dNK infiltration, does not reduce $\alpha 1$ integrin expression.

3) they demonstrate alterations in spiral artery modifications, and demonstrate clearly that they are linked indeed with KK cytotoxicity, but also that a least a major part of it is liked with ANTIBODY DEPENDENT CELLULAR CYTOTOXICITY, not simply cellular cytotoxicity, using anti B3 IgG DnK as a model. To directly prove ADCC involvement, they block ADCC by blocking dNK FC receptore, by anti Fc γ III ani bidy, thus blocking the link between dNK and cytotoxic antibody, the rendering ADCC impossible. This causes a disparition of FNAIT and associated pathologies

Points listed in 1 are NOT new per se. Involvement of NK cells in abortion, IUGR, etc... has been amply demonstrated using several models of immune pregnancy loss, including for a start the CBA x DBA/2 mating combination as well as other strains combination, later on , as well as hyper activation of NK cells, be them dNK of peripheral blood ones, such as dsRNAs (but not only), and this correlated in other models resulting in local dNK activation by other pathways, such as stress. None of these are quoted. For example, they surprisingly quote the Hadda and Duclos paper from the Baines group (1994), but not the seminal 1986 one (de Fougerolles AR, Baines MG. Modulation of the natural killer cell activity in pregnant mice alters the spontaneous abortion rate. J Reprod Immunol. 1987 Jun; 11(2): 147-53.) , nor for TH-17 : Regulation Fu B, Li X, Sun R, Tong X, Ling B, Tian Z, Wei H. Natural killer cells promote immune tolerance by regulating inflammatory TH17 cells at the human maternal-fetal interface. Proc Natl Acad Sci U S A. 2013 Jan 15; 110(3): , nor for antibody action are discussed the effects of anti ICAM-1 and anti LFA-1 as (one) of the alternative examples (Takeshita T, Satomi M, Akira S, Nakagawa Y, Takahashi H, Araki T. Preventive effect of monoclonal antibodies to intercellular adhesion molecule-1 and leukocyte function-associated antigen-1 on murine spontaneous fetal resorption. Am J Reprod Immunol. 2000 Mar; 43(3): 180-5.), nor of course several other papers in the CBA x DBA/2 model. As another example, they quote preeclampsia but fail to mention the work by Girardi and /or Salmon. This is either a big error or very unfair .

Linking of dNK and FNAIT is new, and points 2 and 3 are NEW and Per se make the M/S worth meriting publication.

However, again, when speaking of NK and clinical situation, they do not quote any of the (many) laboratories which treat or "treat" implantation failure , recurrent pregnancy loss by addressing NK activation. Same comment . the sentence "These better outcomes of pregnancy suggest that manipulations of NK cells could be important in the management of FNAIT and other antibody-related pregnancy losses" must be followed by as already suggested by ...

Reviewers' comments:

Reviewer #1 (Remarks to the Author):

The authors have satisfactorily addressed my concerns.

Reviewer #2 (Remarks to the Author):

The authors have addressed our most important concerns with their new and revised experiments, but there are still some points that need to be improved and clarified. Please refer to our list below:

Throughout all the legends - The authors made the effort to improve the description of the panels, but there are still problems. For example - in the legend of figure 4, using "top panel" to indicate figure 4A2 is redundant - what should have been used was "left" and "right" panels to describe the various panels in figure 4A2. The same way, using "bottom panel" for Figure 4B does not help - it does not have top and bottom panels, but left, middle and right panels. Please try to improve the description to make the legends more clear.

Figure 1A1 - For the plot Y-axis label, use "Fetal weight" instead of "Fetuses weight".

Section 3 - The word "underlining" is misused - should be "underlying".

Legend of Figure 3 - there is a reference to NKp46 staining in the legend of figure 3A, but no panels show NKp46.

Figure 3B2 – Since the western blot samples were lysates of whole placenta and not of isolated NK cells, the increased NKp46 is not necessarily an indication of higher expression in NK cells, but of an increased number of NK cells in the tissue. Please correct your interpretation.

Figure 3B2 bottom – There is a large difference in the number of cells acquired between the two flow cytometry plots. This may exacerbate the differences. Also, the plots look like they were not correctly compensated. In addition, the label in the Y axis is confusing – should have only NKp46 and say that cells were gated in DBA+.

Figure 3C -there should be a supplementary figure showing that the depletion was successful.

Figure 6A – "Immune + anti rat IgG" photo is included under the label "Control". Shouldn't it be under "active FNAIT" Also, the rationale for the use of IVIG, moderate vs passive FNAIT and how it was induced should be better described.

Reviewer #3 (Remarks to the Author):

The 2 other referees went into much more detail than I did, and upon re-reading, I agree with their comments ... and I believe the manuscript has been extensively revised according to their requests (and mine).

REVIEWERS' COMMENTS:

Reviewer #1 (mediating for reviewer#2) (Remarks to the Author):

The authors have satisfactorily addressed the reviewer's remaining concerns.

We appreciate the encouraging comments from the Reviewers and the Editorial Board that our study is informative, of considerable potential interest and importance, and thank Dr. Ching-yu Huang for inviting our manuscript entitled: "Activated NK cells cause placental dysfunction and miscarriages in fetal alloimmune thrombocytopenia" (our reference NCOMMS-16-28790-T) for revision. We appreciate the insightful comments from the reviewers, and have performed additional experiments following their recommendations and the guidance of the editorial board. The manuscript has been accordingly revised. We believe that we have addressed the expressed concerns, and the quality of the manuscript has been significantly improved after adding these new data. The changes in the manuscript have been highlighted using red font. Our detailed, point by point, responses to the editors and the reviewers are listed below:

ANSWERS TO REVIEWER COMMENTS:

Reviewer #1 (Remarks to the Authors):

In this report, the authors investigate the role of natural killer (NK) cell signaling in fetal and neonatal alloimmune thrombocytopenia (FNAIT). The experimental approach utilizes an Itgb3 null mouse model transfused with wild type platelets and then bred to wild type males. The wild type platelet transfusions stimulated an immune response in the Itgb3 null females and the development of anti-beta 3 integrin antibodies, which resulted in IUGR and ultimately fetal death at ~E14.5. The experimental work is informative and the results are of considerable interest. Some concerns regarding the experimental design, presentation and interpretation of the data, including previously published reports, are presented below.

Response: We thank the reviewer for the positive comments that our experimental work is informative and the results are of considerable interest. Our responses to reviewer's comments and suggestions are listed below:

1. Abstract and elsewhere in the manuscript: Extravillous trophoblast is a term used to describe a lineage of trophoblast cells from the human placenta. The term is not appropriate for describing trophoblast lineages in the mouse placenta. EVT is used as an abbreviation of extravillous trophoblast and is used inappropriately to describe invasive trophoblast/endovascular trophoblast throughout the manuscript.

Response: We appreciate the reviewer's comments, and have adopted the term invasive trophoblast accordingly thorough the manuscript (manuscript, figures and figure legends).

2. Page 4, paragraph 3, lines 1 and 2: Inappropriate reference for describing interactions between NK cells and trophoblast in the mouse and human. The reference only includes

human data and the findings are based entirely on in vitro experimentation, which is of concern.

Response: We thank the reviewer for the suggestion. We have edited the text and added a reference for murine uterine NK (uNK) cells (page 4 paragraph 3, line 2).

3. Page 7, lines 4-5: The "not shown" data on PGF and sFlt-1 should be presented in the Supplemental figures.

Response: As recommended, we now present data on PGF and sFlt-1 in the Supplemental figures 1 (page 7, lines 4-5).

4. Page 8, paragraph 2: What was the rationale for using biotin transfusion to monitor placental exchange? Biotin is transported by the SMVT protein (Ganapathy et al. J Pharm Exp Therapeutics 294:413-420, 2000). Does the activity of the SMVT transporter reflect all placental transport function? The authors should provide references or describe the logic why monitoring biotin transport across the placenta is an informative method for monitoring placental exchange. Where is the western blotting for biotin transfer data presented?

Response: We thank for the reviewer's recommendation, and have further elaborated the rationale for using biotin to test materno-fetal placental transportation. Commercially available biotin was intravenously injected into pregnant mice after ultrasound, and monitored in both the placentas and fetal tissues to see the amount of endogenous biotin crossing the placenta and reaching the fetal circulation as previously reported (Taniguchi A, Watanabe T 2008: Transplacental transport and tissue distribution of biotin in mice at midgestation), which we think, at least partially, reflect placental transport function. It has been reported that biotinidase and carboxylase activity increased after biotin supplementation. We have provided more details describing our experimental design in page 8, paragraph 2 and page 19 paragraph 1 as well as more references as the reviewer recommended.

During the last submission, we realized that our "Western Blot data" did not add any more information beyond our immunofluorescent staining data (Figure 2C), therefore did not present the data in the last submission, unfortunately we forgot to delete these words, which have now been deleted in the revised manuscript (page8).

5. Page 9, 2nd paragraph: What the rational for using "first trimester" trophoblast cell lines rather than first trimester primary trophoblast cells? The validity and appropriateness of the Htr8 and Swan71 cell lines have been questioned as models for first trimester human trophoblast cells (see Lee et al. What is trophoblast? a combination of criteria define human first-trimester trophoblast. Stem Cell Reports 6:257-272, 2016).

Furthermore expression profiles could be performed on first trimester human trophoblast tissues rather than relying entirely on cell lines.

Response: We appreciated reviewer's comment. As suggested, we performed the recommended experiments for $\beta 3$ integrin expression on primary trophoblasts in human placentas from across early gestation (first trimester of pregnancy). Our data showed expression of $\beta 3$ integrin on extravillous trophoblast (HLA-G positive cells) and syncytiotrophoblast (cytokeratin-7 positive cells). The relevant information has been included in the revised manuscript (Figure 4A1, Page 9).

6. Page 10, 3rd sentence: awkward... a fewer recent dNK infiltration....

Response: We thank the reviewer's comment. We have corrected the third sentence at Page 10, line 6.

7. Page 10, paragraph 2: The results presented seem to contradict the findings of Croy and co-workers, which indicate that NK cells promote uterine spiral artery remodeling, including disappearance of vascular smooth muscle. Consequently, is the conclusion that there are more NK cells, which persist but these NK cells are actually ineffective in their fundamental actions on uterine spiral artery remodeling and instead act primarily to prevent trophoblast-guided uterine spiral artery remodeling? Thus it would appear that

the NK cells persisting in the uterus represent a unique subset of NK cells. Can this subset of NK cells be better defined based on surface marker expression?

Response: We thank the reviewer for these observations and completely agree with the reviewer that this is a very striking finding. As shown in this study the activated uNK cells accumulating in the deciduas of immune mice are effective and harmful for the placenta at the late stage of gestation. The phenotypic change in NKp46 and CD107 expression supports this observation. We further demonstrated that depletion of these activated uNK cells significantly prevented placental pathology and fetal loss. If uNK cells were not effective in their role, we believe that asialo-GM-1 mediated peripheral NK cell depletion and NK cell inhibition would have exacerbated the placental pathology. Our new flow cytometry data (Fig. 3B2) that now incorporates CD107 as a marker of NK cell degranulation/activation show that activated DBA⁺NKp46⁺CD107⁺ uNK cell numbers are significantly increased in immune mice. Further characterization of these unique subsets of uNK cells identified in our studies should be the interesting topic for the future studies.

8. Page 14, 2nd to last sentence: Ref Nos. 31, 53 are inappropriate for the statement. These reports do not indicate that dNK cells prevent/restrain invasion of EVT. Instead these reports describe cooperative roles for dNK cells promoting/facilitating EVT invasion using in vitro models. The best in vivo experimental evidence for NK cells restraining EVT invasion was generated in the rat (Chakraborty et al. PNAS 108:16295-16300, 2011).

Response: We thank the reviewers for their comments. We have deleted references 31 and 53 and cited relevant paper by Chakraborty et al. PNAS 108:16295-16300, 2011, the third sentence at Page 14 in the revised manuscript.

9. Page 17, paragraph one: Using wild type mice purchased from a vendor as a control for mice maintained as a colony is potentially problematic. Mouse colonies maintained at separate facilities show genetic drift overtime and thus introduce additional variables into the experimentation.

Response: We agree with the reviewer that mouse colonies maintained at separate facilities may have genetic drift overtime, and the wild-type mice purchased from a vendor may have introduce additional variables into the experiments. In the present study, the mouse colonies are littermates from $\beta 3$ integrin deficient mouse colonies bred with the wild-type mice purchased from the vendor. Wild-type mice purchased from vendors were used as platelet donors or male breeders.

10. Page 18, paragraph one, last sentence: Some references for the Htr8 and Swan71 cell lines. Additionally how were the cells maintained? How many passages? Do the cells used for the experiments retain hallmarks of 1st trimester trophoblast?

Response: We appreciate the reviewer's comments. As recommended, more details regarding HTR8 and Swan cell lines were introduced. This information is now incorporated in the revised manuscript accordingly (Page 18 line 18). HTR8/SV neo and Swan cells were culture in supplemented RMPI and DMEM F-12K media respectively. Cells were incubated under 5% CO₂ at 37°C. The media were changed every 72h. These cells that were passaged between 14 -20 retained hallmarks of first trophoblasts as shown below by expression of HLA-G by HTR8/SVneo. As recommended for the question 5, we have now also included the primary trophoblasts in human placentas in the studies, which enhanced the quality of our data.

11. Page 19, paragraph one, last two sentences: What is the logic for monitoring biotin transfer? More detail needs to be included for the procedure, especially monitoring biotin in the placenta and fetus. Inclusion of references describing the procedure would be helpful.

Response: We agree and have now elaborated on further details in the procedure as recommended by including new reference (page 8 and 19).

12. Page 20, paragraph two: How was the amount of antibodies used to neutralize NK cells? Is this based on earlier experimental work or were pilot experiments performed to optimize the antibody dosage? Some quantitative data should be included on the effectiveness of the antibodies in decreasing NK cell numbers.

Response: We thank the reviewer for these questions. IVIG treatment is based on our previous study (Chen et al., 2010 Blood and Ni et al., 2006 Blood). Anti- asialo-GM1 and anti-NKp46 (3.33 mg/kg body-weight dosage) were based on manufacture's recommendations. The similar dose for anti-FcγRIIIa (6.66 mg/kg body-weight) was based on a previous publication by Kurlander RJ et colleagues (8 micrograms/g body weight of 2.4G2 by Kurlander RJ et al; J Immunol. 1984 Aug;133(2):855-62). These references are provided in the revised manuscript (Materials and Methods page 20).

13. Page 21, Why DBA negative selection to isolate uterine NK cells? Are the NK cells used in the experiment DBA positive or negative?

Response: We acknowledge the reviewer's concern and regret for mentioning "negative" but not "positive" selection. This has been corrected in the manuscript (page 21). NK cells were retained by DBA lectin-coated beads on magnetic tube holder as previously reported (Croy, B.A., et al. 2010. Analysis of uterine natural killer cells in mice. Methods in molecular biology 612, 465-503). This "positive" selection procedure allowed us to purify DBA positive NK cells.

14. Fig2B: The y-axis for the bar graph is confusing.

Response: We thank the reviewer and we have rectified the labeling. We have corrected the y-axis for the bar graph to be: % of placenta area covered by fetal vessel (IB4 staining) (Fig. 2B).

15. Fig. 2C: What is presented in the bar graph? MFI is not defined.

Response: We thank the reviewer for the suggestion. The streptavidin mean fluorescence intensity (MFI) has now been defined in the Fig. 2C in the revised manuscript.

16. Fig. 3B and elsewhere: MFI is not defined.

Response: We agree and the mean fluorescence intensity (MFI) has been now defined thoroughly in the manuscript and figures (Fig. 3B).

17. Fig. 4B: How were the junctional zone cells identified as glycogen cells? It would seem that the critical NK cell interactions will be with invasive trophoblast.

Response: The junctional zone was identified by the histology. Pictures shown below identify the deciduas, junctional zones and labyrinths. Integrin $\beta 3$ positive cells are predominantly located in the junctional zone. Some invasive trophoblasts also expressed integrin $\beta 3$.

Reviewer #2 (Remarks to the Authors):

This manuscript tests the hypothesis that the miscarriages and intrauterine growth restriction (IUGR) observed in fetal/neonatal alloimmune thrombocytopenia are due to an inflammatory response and placental pathology mediated by decidual NK (dNK) cells, through antibody dependent cell cytotoxicity (ADCC). The authors conclude that antibody production against allogeneic human platelet antigen (HPA)-1a, specifically in integrin $\beta 3$, starts an inflammatory response and mediates ADCC by dNK. Activated dNK are then responsible for trophoblast apoptosis, poor vascular remodeling, and pregnancy failure. The authors present clear in vivo data demonstrating that in the presence of anti-integrin $\beta 3$ antibodies, there is an increase in inflammation, reduction of growth factor production, IUGR and reduced fetal survival. Placental perfusion and vascularization, as well as nutrient transfer to the fetus, are also reduced. The authors also demonstrate an increase in NK cell numbers and granularity, as well as interactions with trophoblast, and show that the phenotype can be reversed by NK cell depletion, blocking of the natural cytotoxicity receptor NKp46, or blocking of Fc receptors. The in vivo mouse data is convincing, but the manuscript has an important flaw – it is not easily translatable to humans. The authors do not show clearly if the effect is due to dNK or to infiltrated peripheral blood NK, which is important since dNK do not express Fc receptors. In addition, the organization of the figures and legends is confusing, and some of the methods are not clear.

Response: We thank the reviewer and really appreciate the positive comments that our in vivo mouse data are convincing. We have performed multiple follow-up experiments to further address these issues.

1. Nomenclature – use uNK when mentioning mice, dNK is for humans. Also, EVT is a human term.

Response: We thank the reviewer for the suggestions, and as recommended, uNK and invasive trophoblast have been used thoroughly in the revised manuscript.

2. “These females lost their heterozygote fetuses around E14.5 and delivered litters of $\beta 3^{-/-}$ offspring that were half the size of litters of non immune mice”. How was this determined? Were the mice sacrificed at E14.5 and the resorptions phenotyped? Or was this just an assumption?

Response: We appreciate reviewer’s comment. The placenta of offsprings from immunized $\beta 3^{-/-}$ female bred with $\beta 3^{-/+}$ mice were phenotyped by immunohistochemistry as showed by representative figures below.

3. The figure legends are insufficient in most cases. The descriptions are not enough to understand what certain plots within panels of the same figure represent. As an example, in figure 3B, the right panel (western blot and bar plot) is not well described. What is “control”? How were this lysates obtained? It would be better to add more letters or to describe the panels (right, left, top bottom) in more detail.

Response: We thank the reviewer for the insightful feedback. As suggested, we have revised the figure legends accordingly, and have further elaborated on the experimental details included. The term **control** has been corrected to normalized NKp46 (Fig. 3B). The placenta tissues were lysed in radioimmuno-precipitation assay (RIPA) buffer, supernatant collected after centrifugation (x10,000g) and stored at -20 °C until Western blotting (details on page 22 in the revised manuscript).

4. The figure organization should also be improved. In many cases, such as figure 5, the panels do not follow the order in which they are described in the manuscript.

Response: As mentioned above we have revised Fig. 5 to follow the order in which they are described in the manuscript and figure legend (page 11 and 26 in the revised manuscript).

5. The main weakness of the manuscript is figure 5B. The authors test the activity of murine NK cells against human cells. This is a xenogeneic interaction. Any mouse antibody that binds a protein expressed in the human cell membrane could induce ADCC in the murine NK cell, since there is no MHC inhibition. This experiment should be scrapped altogether or done using a mouse trophoblast cell line (e.g. SM9-1). In addition, the method used to measure cytotoxicity is caspase-3 fluorescence; a much better method would be a ^{51}Cr release assay.

Response: We share the reviewer’s concerns regarding the possible xenogeneic interaction. Our recent preliminary data regarding ADCC by uNK cells on invasive murine trophoblasts support our previous data regarding human trophoblasts with uNK cells in the presence of anti- $\beta 3$ IgG. This is also supported by earlier observations on cross species studies: (Chantakru S, et al 2002 and contributions from self-renewal and trafficking due to the uterine NK cell population of early pregnancy. 2002. *J Immunol.* ;168(1):22-8). However we regret for being unable to use ^{51}Cr release assay due to the high number of NK cells needed, and also the use radioactivity restricted in our institution.

6. The premise of this manuscript unfortunately does not translate to humans. The authors seek to prove that that the fetal/placental pathology observed in FNAIT is caused by ADCC mediated by uterine NK cells. Several of the experiments do suggest that the inflammation, loss of vascularization, overall placental pathology and fetal loss are NK cell – dependent (NK cell depletion and NKp46 blocking rescue the phenotype) and specifically due to the presence of anti- $\beta 3$ antibodies that lead to ADCC by NK cells (Fc blocking also rescues the phenotype). However, human decidual NK cells do not express Fc receptors. Thus, although the mouse phenotype is convincing, the human disease is probably not dependent on NK cell ADCC. Unless the authors assume that somehow peripheral blood NK cells infiltrate the placenta, but in that case the ADCC is not due to Fc receptors on dNK – it’s pNK-dependent. This is shown by the lack of ADCC by dNK of non-immune mice towards human trophoblast cell lines – these dNK are residents, not recently infiltrated pNK, and do not express Fc receptors. There is also the possibility that the antibody production induces an inflammatory response that leads to uterine NK cell activation and expression of Fc-receptors. In this case, NK cells do not initiate inflammation, but can still be responsible for the placental pathology.

Response: We appreciate the question and fully understand the reviewer’s concern here. We agree that human decidual $\text{CD}56^+\text{CD}16^-$ NK cells do not express the Fc receptor in the normal pregnancy (i.e. the physiological condition). However, an increasing body of evidence suggests that inflammation and chemokines can change the expression of CD16 on dNK cells, and/ or the

migration of CD16 pNK cell migration into to the deciduas in humans and mice, particularly in those pathological scenarios that cause diseases.

A recent study has shown that women with increased numbers of CD16⁽⁺⁾ uNK cells in the deciduas may be at greater risk of developing infertility disorders (Giuliani E et al., 2014.) Characterization of uterine NK cells in women with infertility or recurrent pregnancy loss and associated endometriosis. *Am J Reprod Immunol.* 72(3):262-9). Furthermore, Farghali and colleagues reported an increased CD56⁺ CD16⁺ uNK cells in decidual specimens of women (85%, 68 out of 80 women) with unexplained repeated miscarriages (Farghali et al., 2015 Relationship between uterine natural killer cells and unexplained repeated miscarriage. *J Turk Ger Gynecol Assoc.;*16(4):214-8). In addition it has been reported that peripheral blood NK cell recruitment to the uterus contributes to the accumulation of NK cells during early pregnancy (Carlino C et al. Recruitment of circulating NK cells through decidual tissues: a possible mechanism controlling NK cell accumulation in the uterus during early pregnancy. 2008 *Blood.* 111(6):3108-15). These findings support the idea that dNK cells expressing FcγR may be involved in ADCC in human FNAIT, which is also consistent with the hypotheses listed by the reviewer.

Furthermore, work from Dr. Croy's group also highlighted the importance of FcRgamma receptors in uNK cell activation during spiral artery remodeling (Xie X et al, *Biol Reprod.* 2005 Pathways participating in activation of mouse uterine natural killer cells during pregnancy Sep;73(3):510-8). In the implantation sites of FcRgamma^{-/-}/CD3zeta^{-/-} and FcRgamma^{-/-}/DAP12^{-/-} mice, differentiated, but not functionally impaired uNK cells could not modify spiral arteries. These observations point out that the activated subsets of murine uNK cells and human dNK cells can mediate cytotoxicity against target cells including trophoblast during placenta inflammation as we observed in FNAIT. As mentioned by the reviewer and also in agreement with our observations regarding that infiltration of peripheral CD56⁺CD16⁺ NK cells into decidua, it is likely that ADCC can be mediated by these cells. This information has been discussed in the manuscript and the related references are cited in the revised manuscript (page 15).

7. To identify the origin of the increased NK cell population in “immune” mice, I suggest a procedure similar to the one used in Chiossone et al (2014) *Haematologica* 99(3) 448-457. Peripheral blood NK cells from EGFP+ mice were adoptively transferred to pregnant mice and tracked within the recipient mice. This way the authors can conclude whether the cytotoxic NK cells are activated dNK or infiltrated pNK that express Fc receptors.

Response: We highly appreciate the reviewer's suggestions and share his point of view. We have performed the following experiments to address the questions regarding the origin of placental NK cells that caused pathology. As shown in this figure purified peripheral NK cells infiltrated the placenta at E14.5 which is in agreement with our observation that asialo-GM-1 also depleted peripheral NK cells and ameliorated FNAIT. In addition we showed in this study that non-proliferating NK cells in the deciduas may be coming from circulation (page 10 in the revised manuscript).

Reviewer #3 (Remarks to the Author):

The paper by Yougbaré et al demonstrate in my opinion quite clearly that fetal / Neonatal alloimmune thrombocytopenia (FNAIT) involves decidual NK (cytotoxic) activation and persistence of these cells.

1) They demonstrate clearly that in a murine FNAIT model, the alloimmune thrombocytopenia is correlated with placental and neonatal pathologies, including lower foeto placental weight (Intrauterine growth restriction, IUGR), enhanced abortion /resorbtion rates, placental and decidual bed vascular anomalies (improper development of spiral arteries), a Th-17 bias, anomalies of Sflt-1 and angiogenic cytokine production. This was confirmed by Doppler assessment of placental vascular flow, all that confirmed by (gross ! very gross) anatomical examination of the placenta, and functional studies of placental exchange functions by biotin transfer test. This pathology correlates with an INCREASED decidual bed, itself filled by activated decidual NK cells (dNK), with in case of pathology high NKP46 and perforin expression, indicating an activation towards the cytotoxic pathway rather than engagement in the activation of the trophic and angiogenic pathway. Inhibition of NKP46 or depletion of the excess NK cells by Asialo GM1 reduced this pathology.

2) The NK accumulation and persistence correlates with expression of $\alpha 1$ dNK integrin, nicely suggesting an infiltration process. However, anti Asialo GM1 antibody, while reducing dNK infiltration, does not reduce $\alpha 1$ integrin expression.

3) they demonstrate alterations in spiral artery modifications, and demonstrate clearly that they are linked indeed with KK cytotoxicity, but also that a least a major part of it is linked with ANTIBODY DEPENDENT CELLULAR CYTOTOXICITY, not simply cellular cytotoxicity, using anti B3 IgG DnK as a model. To directly prove ADCC involvement, they block ADCC by blocking dNK FC receptore, by anti Fc γ III ani bidy, thus blocking the link between dNK and cytotoxic antibody, the rendering ADCC impossible. This causes a disparition of FNAIT and associated pathologies.

Response: We highly appreciate the encouraging comments of the reviewer that support our newly described mechanism in FNAIT. We have made several revisions to further address reviewer's concerns. These revisions have overall significantly improved the quality and content of the manuscript.

Points listed in 1 are NOT new per se. Involvement of NK cells in abortion, IUGR, etc... has been amply demonstrated using several models of immune pregnancy loss, including for a start the CBA x DBA/2 mating combination as well as other strains combination, later on, as well as hyper activation of NK cells, be them dNK of peripheral blood ones, such as dsRNAs (but not only), and this correlated in other models resulting in local dNK activation by other pathways, such as stress. None of these are quoted. For example, they surprisingly quote the Hadda and Duclos paper from the Baines group (1994), but not the seminal 1986 one (de Fougerolles AR, Baines MG. Modulation of the natural killer cell activity in pregnant mice alters the spontaneous abortion rate. *J Reprod Immunol.* 1987 Jun;11(2):147-53.) , nor for TH-17 : Regulation Fu B, Li X, Sun R, Tong X, Ling B, Tian Z, Wei H. Natural killer cells promote immune tolerance by regulating inflammatory TH17 cells at the human maternal-fetal interface. *Proc Natl Acad Sci U S A.* 2013 Jan 15;110(3): nor for antibody action are discussed the effects of anti ICAM-1 and anti LFA-1 as (one) of the alternative examples (Takeshita T, Satomi M, Akira S, Nakagawa Y, Takahashi H, Araki T. Preventive effect of monoclonal antibodies to intercellular adhesion molecule-1 and leukocyte function-associate antigen-1 on murine spontaneous fetal resorption. *Am J Reprod Immunol.* 2000 Mar;43(3):180-5.), nor of course several other papers in the CBA x DBA/2 model. As another example, they quote preeclampsia but fail to mention the work by Girardi and /or Salmon. This is either a big error or very unfair.

Linking of dNK and FNAIT is new, and points 2 and 3 are NEW and Per se make the M/S worth meriting publication. However, again, when speaking of NK and clinical situation, they do not quote any of the (many) laboratories which treat or “treat” implantation failure, recurrent pregnancy loss by addressing NK activation. Same comment the sentence “These better outcomes of pregnancy suggest that manipulations of NK cells could be important in the management of FNAIT and other antibody-related pregnancy losses” must be followed by as already suggested by .

Response: We thank the reviewer for the feedback and understand their concern. We regret that the allowance of only 70 references for the first submission did not permit us to cite and acknowledge these earlier studies. We have now revised the manuscript and cited these work accordingly in the introduction (we cited at **Page 4**, de Fougerolles R, Baines MG. Modulation of the natural killer cell activity in pregnant mice alters the spontaneous abortion rate. *J Reprod Immunol.* 1987 Jun;11(2):147-53 and Fu B, Li X, Sun R, Tong X, Ling B, Tian Z, Wei H. Natural killer cells promote immune tolerance by regulating inflammatory TH17 cells at the human maternal-fetal interface. 2013. *Proc Natl Acad Sci U S A.* 15;110(3):E231-40.) and the discussion (we cited at page,1 4, Girardi G1, Complement activation induces dysregulation of angiogenic factors and causes fetal rejection and growth restriction. *J Exp Med.* 2006 Sep 4;203(9):2165-75), and these changes significantly improve the quality and content of the manuscript.

Thank you for the positive feedback regarding our manuscript entitled “Activated NK cells cause placental dysfunction and miscarriages in fetal alloimmune thrombocytopenia (reference NCOMMS-16-28790A)”. We are glad to hear that the reviewers feel that we have satisfactorily addressed their concerns. The changes in the manuscript have been highlighted using red font. Our detailed, point by point responses to Reviewer #2 (both review#1 and #3 have been satisfied with our last revision) are listed below:

Reviewer #1 (Remarks to the Author): The authors have satisfactorily addressed my concerns.

Reviewer #2 (Remarks to the Author):

The authors have addressed our most important concerns with their new and revised experiments, but there are still some points that need to be improved and clarified. Please refer to our list below:

Throughout all the legends - The authors made the effort to improve the description of the panels, but there are still problems. For example - in the legend of figure 4, using "top panel" to indicate figure 4A2 is redundant - what should have been used was "left" and "right" panels to describe the various panels in figure 4A2. The same way, using "bottom panel" for Figure 4B does not help - it does not have top and bottom panels, but left, middle and right panels. Please try to improve the description to make the legends more clear.

Response: We thank the reviewer for the positive comments and also for appreciating our effort in thoroughly revising the manuscript and figures. Changes to the legends of figure 4 (A2 and 4B) have been made accordingly, on pages 26 and 27 of the revised manuscript.

Figure 1A1 -For the plot Y-axis label, use "Fetal weight" instead of "Fetuses weight".

Response: We thank the reviewer for the comment and have made correction regarding fetal weight (Figure 1A1).

Section 3 - The word "underlining" is misused - should be "underlying".

Response: We thank the reviewer and regret misusing the term "underlining". We have now used the proper term "underlying" in page 9, first sentence.

Legend of Figure 3 - there is a reference to NKp46 staining in the legend of figure 3A, but no panels show NKp46.

Response: We have edited and unified the legend. 'NKp46 staining' in figure 3A has been removed (page 26).

Figure 3B2 – Since the western blot samples were lysates of whole placenta and not of isolated NK cells, the increased NKp46 is not necessarily an indication of higher expression in NK cells, but of an increased number of NK cells in the tissue. Please correct your interpretation.

Response: We agree with the reviewer that it could be an increased of NK cells, however we could not exclude the possibility of the upregulation of NKp46 expression on NK cells since this receptor is involved in activation and granule release. Moreover, we have demonstrated an increased of NKp46 expression on purified uNK cells in Fig. 5B.

Figure 3B2 bottom – There is a large difference in the number of cells acquired between the two flow cytometry plots. This may exacerbate the differences. Also, the plots look like they were not correctly compensated. In addition, the label in the Y axis is confusing – should have only NKp46 and say that cells were gated in DBA+.

Response: We acknowledge the reviewer's comments and share the concern expressed. The NK cells isolated from the same number of placenta (then pooled together) were lower in naive mice compared to immune mice. This is consistent with the data showed on figure 3B1 for increased uNK cells in immune. For that reason we also observe on the plot the same trends of lower number of NK cell number in naive compared to immune mice. Our experimental design using multicolor flow cytometry and fluor-conjugated antibodies that avoid photobleaching between channels (DBA-Alexa488, NKp46-Vioblue and CD107-APC) helped us to properly gate and compensate activated uNK cells. The bar graph uses percentage rather than cell number, which account for the differences in cell number.

We have removed DBA⁺ the Y axis labelling (Figure 3B2 bottom).

Figure 3C -there should be a supplementary figure showing that the depletion was successful.

Response: We thank the reviewer for the suggestion. We have provided figure for uterine NK cell depletion (figure 4D) which demonstrated decreased NK cell accumulation in the placenta. Previous studies have already shown that anti-asialo-GM-1 antibody mediated peripheral NK cells depletion (Lala PK1, Scodras JM, Graham CH, Lysiak JJ, Parhar RS. Activation of maternal killer cells in the pregnant uterus with chronic indomethacin therapy, IL-2 therapy, or a combination therapy is associated with embryonic demise. Cell Immunol. 1990 May;127(2):368-81 and Chen D, Weber M, Lechler R, Dorling A. NK-cell-dependent acute xenograft rejection in

the mouse heart-to-rat model. Xenotransplantation. 2006 Sep;13(5):408-14.). Therefore we have shown that the depletion of uNK cells was successful in placenta.

Figure 6A – "Immune + anti rat IgG" photo is included under the label "Control". Shouldn't it be under "active FNAIT" Also, the rationale for the use of IVIG, moderate vs passive FNAIT and how it was induced should be better described.

Response: We thank the reviewer for the comment. We have edited figure 6A. "Control" and "active FNAIT" were removed from figure 6. We have provided more rationale for using IVIG, moderate and passive FNAIT.

Rationale:

- IVIG is an important therapy in FNAIT management was compared to anti-NK therapies (page 21).
- To compare severity of FNAIT and efficacy of anti-NKp46 treatment we used different number of WT platelets (1×10^8 for severe FNAIT or 1×10^7 for moderate FNAIT) incorporate in page 17.
- Injections of anti- $\beta 3$ *immune* sera into non-immune mice (bred by WT male) served as a passive FNAIT model to test whether anti- $\beta 3$ alone can induce or not ADCC (incorporate in page 17).

Reviewer #3 (Remarks to the Author):

The 2 other referees went into much more detail than I did, and upon re-reading, I agree with their comments ... and I believe the manuscript has been extensively revised according to their requests (and mine).